# Soluble E-cadherin promotes tumor angiogenesis and localizes to exosome surface

Maggie K.S. Tang[1], Patrick Y.K. Yue[2], Philip P. Ip [3], Rui-Lan Huang [4], Hung-Cheng Lai[4], Annie N.Y. Cheung[3], Ka Yu Tse[5], Hextan Y.S. Ngan[5] & Alice S.T. Wong[1]

The limitations of current anti-angiogenic therapies necessitate other targets with complimentary mechanisms. Here, we show for the first time that soluble E-cadherin (sE-cad) (an 80-kDa soluble form), which is highly expressed in the malignant ascites of ovarian cancer patients, is a potent inducer of angiogenesis. In addition to ectodomain shedding, we provide further evidence that sE-cad is abundantly released in the form of exosomes. Mechanistically, sE-cad-positive exosomes heterodimerize with VE-cadherin on endothelial cells and transduce a novel sequential activation of β-catenin and NFκB signaling. In vivo and clinical data prove the relevance of sE-cad-positive exosomes for malignant ascites formation and widespread peritoneal dissemination. These data advance our understanding of the molecular regulation of angiogenesis in ovarian cancer and support the therapeutic potential of targeting sE-cad. The exosomal release of sE-cad, which represents a common route for externalization in ovarian cancer, could potentially be biomarkers for diagnosis and prognosis.

[1] School of Biological Sciences, University of Hong Kong, Pokfulam Road, Pokfulam, Hong Kong. [2] Department of Biology, Hong Kong Baptist University, Kowloon Tong, Hong Kong. [3] Department of Pathology, University of Hong Kong, Sassoon Road, Pokfulam, Hong Kong. [4] Department of Obstetrics and Gynecology, Shuang-Ho Hospital, Taipei Medical University, Taipei, Taiwan. [5] Department of Obstetrics and Gynecology, University of Hong Kong, Sassoon Road, Pokfulam, Hong Kong. These authors contributed equally: Patrick Y. K. Yue, Philip P. Ip. Correspondence and requests for materials should be addressed to A.S.T.W. (email: awong1@hku.hk)

Tumor vasculature is an attractive therapeutic target. The fact that both the progressive growth of ovarian cancer and the formation of malignant ascites are critically dependent on angiogenesis suggests that anti-angiogenic therapeutic strategies may be meritorious to ovarian cancer treatment[1,2]. In particular, the burden of ascites as a complication in malignancy remains vitally important, as the extent of which is a characteristic of the aggressiveness and metastatic potential and a significant indicator of poor prognosis. Current peritoneocentesis is not effective in addressing the root cause of fluid accumulation and posing a significant risk of side effect[3]. Due to its central role in tumor angiogenesis, vascular endothelial growth factor (VEGF) has emerged as the most important angiogenic target. VEGF is overexpressed in most ovarian cancers. However, despite early clinical benefits in which VEGF has been targeted, most patients ultimately experience the development of resistance and disease progression, suggesting that other angiogenic regulators with complimentary mechanisms are needed[4].

One of the hallmarks of metastatic progression is the dynamic regulation of cadherins (major cell–cell adhesion molecules) that play crucial roles in various aspects of the process, including cell growth, invasion, and migration[5]. Although E-cadherin is synthesized as a transmembrane molecule (a 120 kDa glycoprotein), it can be cleaved off the ectodomain and released in a soluble form (sE-cad; 80-kDa), and this accounts for the decreased expression of functional E-cadherin at the cell surface[6]. This has been largely overlooked in the past because sE-cad can only be detected by examining protein size on western blots. Importantly, sE-cad is highly expressed in the serum and ascites of ovarian cancer (6.18–89.56 μg mL$^{-1}$) and predicts a poor prognosis[7]. These observations underscore the importance of understanding the role of sE-cad in ovarian cancer. In general, sE-cad has only been considered in weakening cell–cell adhesion[8]. There is no information on whether sE-cad also has biological function itself which is critical for dictating metastatic spread. Moreover, the release of sE-cad has only been characterized in the mechanism of ectodomain shedding. While sE-cad has been found to be arisen from the tumor itself[9], it is unclear whether there is other cleavage event.

Here, we show for the first time, in vitro and in vivo, that sE-cad is a pivotal regulator of angiogenesis. We also provide evidence that exosomes are a novel major platform for the cleavage and release of sE-cad in this process.

## Results

**sE-cad promotes HUVEC angiogenesis.** Angiogenesis involves multiple steps, which include the disruption of the vasculature, cell migration, proliferation, and tube formation[10]. As such, we assayed for these activities to understand the mechanism of action. We first examined the endogenous level of sE-cad in three different human ovarian cancer cell lines (OVCAR-3, Caov-3, and OV-90). OVCAR-3, which possesses less or no metastatic potential, showed little expression of sE-cad protein, whereas the protein was highly expressed in Caov-3 and OV-90, which have been shown to frequently metastasize when inoculated in mice[11] (Supplementary Fig. 1a). The results also showed no sN-cad or sP-cad expression, except Caov-3, which also had high sP-cad content (Supplementary Fig. 1a). sE-cad, sN-cad, and sP-cad were absent in normal human ovarian surface epithelial (OSE) and fallopian tube epithelial (FTE) cells (Supplementary Fig. 1a, b). Caov-3 and OV-90, which had the highest sE-cad content of these cell lines analyzed, were used in subsequent experiments.

As shown, sE-cad was a potent stimulant of migration for human umbilical vein endothelial cells (HUVECs) (Fig. 1a). The use of an E-cadherin-neutralizing antibody against the

ectodomain of E-cadherin HECD-1 to immunodeplete sE-cad from the conditioned media (Supplementary Fig. 2) resulted in diminished migration, confirming that the effect was sE-cad-specific (Fig. 1a). Isotype-matched mouse IgG-treated sE-cad had no effect (Fig. 1a). A recombinant Fc/sE-cad chimera was migratory for HUVECs in the μg mL$^{-1}$ range, equivalent to the concentration present in the ascites (Supplementary Fig. 3a). Fc alone had no effect (Supplementary Fig. 3a). Neither conditioned media (Fig. 1b) nor Fc/sE-cad (Supplementary Fig. 3b) induced mitogenesis of HUVECs using the 3-(4,5-dimethylthiazol-2-yl)-2,5-diphenyltetrazolium bromide-based cell viability assay, suggesting that sE-cad may not act as an endothelial cell mitogen. The breakdown of the vascular barrier has an important role in ascites formation and enhances metastasis[12]. To examine whether sE-cad induces endothelial barrier dysfunction, we performed in vitro permeability assay. Whereas under nonstimulated conditions, FITC–dextran flux did not occur across the monolayer, treatment of conditioned media (Fig. 1c) or Fc/sE-cad (Supplementary Fig. 3c) induced significant FITC–dextran flux. These results suggest that sE-cad mediates vascular permeability.

We next investigated the ability of sE-cad to promote the formation of three-dimensional capillary-like tubular structures of HUVECs on the basement membrane matrix mimics Matrigel, which encompasses all steps of angiogenesis[13]. As shown, conditioned media (Fig. 1d) or Fc/sE-cad (Supplementary Fig. 3d) caused a significant increase in tube formation, mimicking a physiological vasculature. These angiogenic effects were completely reversed by the addition of HECD-1-blocking antibodies, but not control IgG, indicating a critical role for sE-cad in the angiogenic phenotypes. Similar results were observed in human microvascular endothelial cells (HMVECs), which cover different parts of the vasculature, indicating that these effects were not restricted to HUVECs (Supplementary Fig. 4). In contrast, immunodepletion of sP-cad from the conditioned media of Caov-3, whereby sP-cad was most abundant, had no effect at all, showing that the angiogenic effect is specifically dependent on sE-cad (Supplementary Fig. 5a, b). Consistent with previous observation[14], Caov-3 and OV-90 expressed little or no VEGF. The VEGF-specific band could only be detected in concentrate conditioned media (30-fold) at the pg mL$^{-1}$ range (Supplementary Fig. 5c). VEGF could be selectively removed from the conditioned media by immunodepletion with the anti-VEGF antibody (Supplementary Fig. 5d)[15]. The addition of anti-VEGF, at a concentration of 10 μg mL$^{-1}$, which was shown to completely block the mitogenic activity of 10 ng mL$^{-1}$ VEGF in HUVECs[16], did not cause significant alterations of migration and tube formation (Supplementary Fig. 5e, f). It is however possible that, in vivo, VEGF concentrations might reach higher levels due to diffusion constrains within intact tissues[17]. It has also been shown that extracellular processing by proteases is required to activate VEGF release[16].

To determine whether sE-cad was angiogenic in vivo, we employed a Matrigel implant murine model. This model has been standardized extensively for studying morphological and functional neovascularization[18]. As shown in Fig. 1e and Supplementary Fig. 3e, Matrigel containing conditioned media or Fc/sE-cad revealed extensive angiogenesis in the implants that were corroborated with the in vitro results. The presence of intact red blood cells inside the neovessels indicated that they are functional (Fig. 1e and Supplementary Fig. 3e). In contrast, treatment of HECD-1 inhibited the sE-cad-induced neovascularization. Thus, sE-cad also has the capacity to induce angiogenesis in vivo.

**sE-cad is secreted from ovarian cancer cells in exosomes.** For both Caov-3 and OV-90, the presence of the 80 kDa E-cadherin

fragment corresponding to sE-cad production in the cytoplasm suggested that sE-cad released was composed of two forms: a membrane-cleaved sE-cad form and a full-length form in membrane vesicles (Fig. 2a). To enable a refined analysis of sE-cad release, we used sucrose gradient centrifugation. Interestingly, while E-cadherin was detected in the plasma membrane-enriched fractions in the bottom of the gradient, most (~70%) E-cadherin was predominantly expressed in Golgi/Trans-Golgi network-enriched fractions where E-cadherin was present and colocalized

with the Golgi marker GM130 (Fig. 2b). The cytoplasmic fragment of E-cadherin (38 kDa), possibly generated by cleavage of the full-length E-cadherin, was also observed (Fig. 2b). These results suggest that E-cadherin ectodomain cleavage occurs not only at the plasma membrane, but also in the Golgi/Trans-Golgi network.

Because a significant amount of sE-cad is found in the Golgi/trans-Golgi network, we postulated that it might exist in microvesicles[19]. To test this assumption, the isolated vesicles

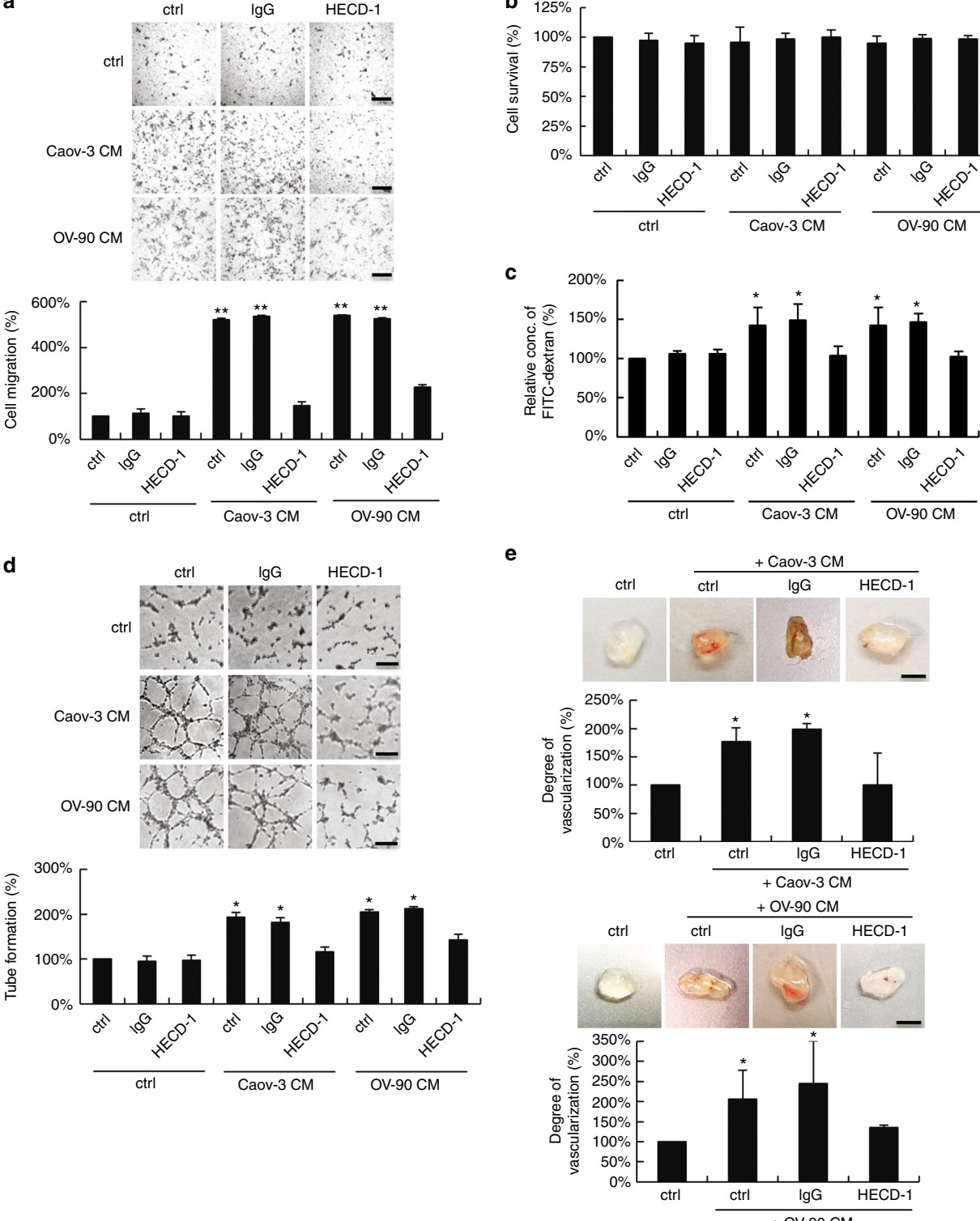

were analyzed with other exosomes markers, such as the tetraspan CD63 and Tsg101, that were also present in the isolated vesicles (Fig. 2c). To exclude potential interference of the subcellular fractions, we performed analogous experiments with GRP78 (endoplasmic reticulum marker), cytochrome c (mitochondria marker), and nucleoporin p62 (nuclear envelope marker) that confirmed their absence in the isolated fractions of the gradient (Fig. 2c). As revealed by sucrose density centrifugation, sE-cad containing vesicles floated in the middle of the gradient of density 1.1036–1.1612 g cm$^{-3}$ positive for Hsp70, an established marker for exosomes (Fig. 2d). Electron microscopy analysis of the sucrose step gradient fractions identified disc-shaped vesicles with a diameter of around 100 nm in the fractions, typical of exosomes (50–150 nm)[20] (Fig. 2e). Localization of sE-cad was also documented by immunoelectron microscopy on the exosomal surface (Fig. 2e). CD63, commonly identified on the surface, was included as a positive control (Fig. 2e). No staining was detected on exosomes stained with control antibody (Fig. 2e). Nanoparticle tracking analysis was performed using ZetaView and revealed $2.4 \times 10^7$ particles per µg of Caov-3 exosomes with a median diameter of 117.8 nm and $4.1 \times 10^7$ particles per µg of OV-90 exosomes with a median diameter of 122.8 nm (Fig. 2f).

**sE-cad-positive exosomes promote angiogenesis.** Next, we investigated the putative angiogenic activity of sE-cad-positive exosomes. As shown in Fig. 3, sE-cad-positive exosomes were a potent inducer of angiogenesis. The enhanced endothelial cell migration (Fig. 3a), permeability (Fig. 3b), and tube formation (Fig. 3c), as well as neovascularization in Matrigel implants (Fig. 3d), were blocked in the presence of mAb to E-cadherin. These results are in agreement with above studies, which showed that expression of sE-cad in conditioned media or Fc/sE-cad was sufficient to induce angiogenesis. Next, sE-cad-positive exosomes were labeled with a BODIPY-TR ceramide fluorescent dye. We showed that the labeling occurred on HUVECs, which could be reverted by treatment with HECD-1, suggesting the uptake of exogenous purified sE-cad-positive exosomes by HUVECs (Fig. 3e). In contrast, no labeling was observed in the control sample (Fig. 3e). Interestingly, in line with a dose-dependent angiogenic response as shown in conditioned media (Supplementary Fig. 6a) or Fc/sE-cad (Supplementary Fig. 6b), OV-90, which expresses high levels of sE-cad-positive exosomes (Supplementary Fig. 6c), showed much increased angiogenic responses than in Caov-3 (Fig. 3d). These results were corroborated with HUVECs treated with sE-cad-positive exosomes at doses 10–50 µg mL$^{-1}$, which caused a significant increase in endothelial tube formation (Supplementary Fig. 6d). In addition, sE-cad-positive exosomes demonstrated a similar angiogenic response in endothelial cell migration and tube formation even at a protein concentration four times lower than sE-cad proteins from

conditioned media (Supplementary Fig. 7). sE-cad-positive exosomes also gave an angiogenic effect similar to that of a clinically relevant dose (10 ng mL$^{-1}$) of VEGF in malignant ascites (Supplementary Fig. 8a)[21], demonstrating potent angiogenic efficacy of sE-cad-positive exosomes. In addition, a specific blocking antibody against VEGF shown to block VEGF-induced endothelial tube formation had no effect on sE-cad-positive exosome-mediated tube formation by these HUVECs (Supplementary Fig. 8a). To further investigate any role of sE-cad in regulating VEGF, we examined the effect of sE-cad-positive exosomes on VEGF expression. As shown in Supplementary Fig. 8b, sE-cad-positive exosomes showed no effect on VEGF expression. Moreover, when combined, sub-threshold doses of sE-cad-positive exosomes and VEGF significantly promoted endothelial tube formation (1.4-fold) ($P < 0.05$), at which neither dose of sE-cad-positive exosomes nor VEGF alone significantly affected endothelial tube formation (Supplementary Fig. 8c), consistent with the VEGF independence that we observed.

**VE-cadherin as sE-cad binding protein.** Because HUVECs do not express E-cadherin, other cell surface molecule may be involved. For this, we carried out affinity chromatography using His-tagged Fc/sE-cad chimera on Ni-NTA beads. We identified VE-cadherin as a putative sE-cad binding protein (but not other cadherins or integrins) on HUVEC surface (Fig. 4a). To obtain evidence for the direct binding of sE-cad to VE-cadherin, HUVECs were treated with Caov-3-derived exosomes, immunoprecipitated with anti-VE-cadherin, and then subjected to immunoblotting with anti-E-cadherin and anti-VE-cadherin antibodies. As shown in Fig. 4b, cell surface-bound E-cadherin was readily detected. In the presence of anti-VE-cadherin, the effect of sE-cad-positive exosomes on angiogenesis by enhancing endothelial cell tube formation was clearly abrogated (Fig. 4c), indicating that sE-cad acts through a VE-cadherin-dependent mechanism and induces angiogenesis.

**sE-cad-positive exosomes mediate angiogenesis via β-catenin.** β-catenin and p120 catenin associate with the intracellular domain of cadherins, promote both cell–cell adhesion and cell signaling[22]. The most important aspect of these catenins is the balance between its cadherin-bound forms and the nuclear pool; only nuclear β-catenin and p120 catenin can affect cell signaling activity. We did western blot analysis to examine the subcellular distribution of β-catenin and p120 catenin in HUVEC in response to exosomes, and found that β-catenin accumulated extensively in the nuclei upon 30 min stimulation and was maintained even 60 min later (Fig. 4d). The amounts of p120 catenin protein estimated by western blotting were the same (Fig. 4d). Nuclear β-catenin was specific, as the signal was lost after HECD-1 mediated inhibition of sE-cad (Fig. 4e). Importantly, β-catenin knockdown significantly inhibited angiogenic tube formation in the presence of sE-cad-positive exosomes

**Fig. 1** sE-cad promotes angiogenesis in vitro and in vivo. **a** Cell migration assay of HUVEC treated with control (ctrl) or immunodepleted conditioned medium (CM) of Caov-3 and OV-90. Upper: Representative images of HUVEC migration. Lower: Quantification of the percentage change of the number of migrated cells. Bar, 100 µm. **b** Proliferation assay of HUVEC treated with control (ctrl) or immunodepleted conditioned medium (CM) of Caov-3 and OV-90 in the presence or absence of E-cadherin neutralizing antibodies, HECD-1 (100 µg mL$^{-1}$). **c** Permeability analysis of HUVEC measured by the percentage change of FITC–dextran flux (excitation 485 nm, emission 535 nm) treated with control (ctrl) or immunodepleted conditioned medium (CM) of Caov-3 and OV-90. **d** Tube formation assay of HUVEC treated with control (ctrl) or immunodepleted conditioned medium (CM) of Caov-3 and OV-90. Upper: Representative images of HUVEC tube formation assay. Lower: Quantification of the percentage change of the number of branching points. Bar, 100 µm. **e** In vivo Matrigel plug implant model using C57/BL6 mice subcutaneously injected with control (ctrl) or immunodepleted conditioned medium (CM) of Caov-3 and OV-90. In vivo neovascularization is measured by the Drabkin's reagent kit after 7 days. Upper: Representative images of excised Matrigel plug. Lower: Quantification of the percentage change in hemoglobin content. Bar, 5 mm. For in vivo Matrigel plug model, $n = 6$ per group, and were conducted twice. For the other assays, $n = 3$ per group, all experiments were repeated three times. Error bar indicates SD of the mean. *$P < 0.05$, **$P < 0.01$ versus untreated control using one-way analysis of variance followed by Tukey's least significant difference post hoc test

(Fig. 4f). Treatment of HUVEC with anti-VE-cadherin showed similar, much inhibited nuclear β-catenin responses to sE-cad-positive exosomes (Fig. 4g), suggesting that β-catenin is critically involved in the sE-cad/VE-cadherin-mediated heterodimerization that contributes to increased angiogenesis.

**sE-cad-positive exosomes activate the NFκB signaling cascade.** To investigate novel signaling paths downstream of sE-cad–VE-cadherin in HUVECs, we compared sE-cad-positive exosomes-treated HUVECs cultured with or without HECD-1 using Affymetrix gene chips Human Gene 2.0 ST, which covers over 30,000

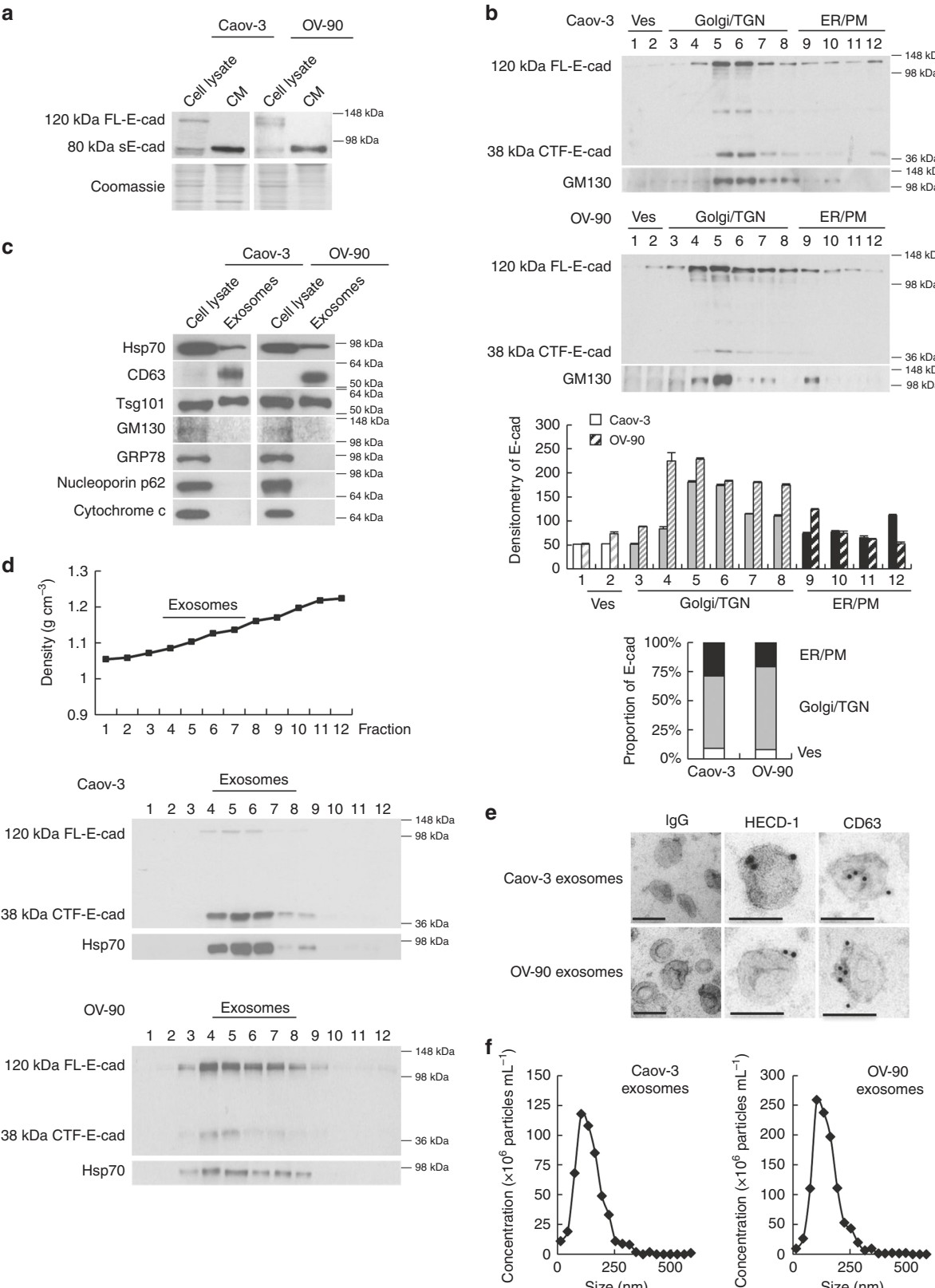

coding transcripts. 840 different genes were significantly upregulated and 691 were significantly downregulated by at least 1.5-fold (Fig. 5a). The genes were subjected to ingenuity pathway analysis (IPA), in which NFκB is in the central position of these networks (Fig. 5a). The altered transcriptional activities of NFκB in response to sE-cad-positive exosomes and the interaction with VE-cadherin were confirmed using luciferase reporter assays (Fig. 5b). NFκB activation cascade is characterized by the nuclear translocation of p65/p50 or RelB/p52 subunits[23]. Western blot analysis of the nuclear extract of HUVECs identified p65/p50 as the key player with sE-cad-positive exosomes (Fig. 5c). The NFκB inhibitor, Bay11-7082, potently inhibited angiogenic tube formation in the presence of sE-cad-positive exosomes (Fig. 5d). These results indicate that NFκB contributes to the angiogenic behavior of sE-cad-positive exosomes and suggest that activity could be mediated through the classical NFκB pathway. Several studies provide evidence for interactions between the β-catenin and NFκB signaling pathways[23–25]. By using coimmunoprecipitation studies, we did not detect β-catenin in the p65 coimmunoprecipitated complex nor in the reciprocal experiment in both control and sE-cad-positive exosome-treated HUVECs (Fig. 5e). Moreover, knocking down β-catenin by siRNA had no effect on NFκB-dependent transcriptional activities (Fig. 5f). Nor did p105/p50 and p65 siRNA affect β-catenin/TCF-dependent transcriptional activities (Fig. 5f). Interestingly, as we followed the time course of the status of these two pathways, we found that β-catenin was activated and already at its maximum by 30 min. On the contrary, nuclear translocation of NFκB was not noted until 120 min (Fig. 5g). The downregulation of either β-catenin or NFκB by siRNA could abolish the sE-cad-positive exosomes-induced angiogenic tube formation, whereas the inhibition of both β-catenin and NFκB had no additive effects (Fig. 5h). These data suggest that these two pathways may signal through sequential and independent cascades.

**sE-cad-positive exosomes trigger angiogenesis in vivo**. To determine whether the in vitro effect on angiogenesis was recapitulated in vivo, sE-cad-negative HEYA8 (Supplementary Fig. 1c) tumor-bearing mice were treated intraperitoneally twice weekly with IgG or HECD-1 pretreated OV-90-derived exosomes for 21 days. OV-90-derived exosomes induced a dose-dependent increase in the number of disseminated tumor nodules and ascites formation, which was noted at 5 μg and significant at 15 and 25 μg per dose (Supplementary Fig. 6e). Treatment with HECD-1 significantly delayed both the number of disseminated tumor nodules and ascites formation compared with IgG control mice (Fig. 6a). In addition, HECD-1 treatment led to a marked decrease in neovascularization, which was characterized by newly formed immature vessels, indicating a role for sE-cad-positive exosomes in ovarian tumor angiogenesis (Fig. 6b). CD31 staining also revealed a decrease in tumor microvessel density (MVD) (Fig. 6b). Additionally, sE-cad-positive exosomes produced a marked increase in extravasation of Evans blue, which indicates albumin and plasma protein leakage into the interstitial tissue[26] (Fig. 6c). Treatment with HECD-1 resulted in a significant

inhibition of tumor vessel leakage (Fig. 6c). Next, we isolated vesicles from the ascitic fluids of ovarian carcinoma patients ($n = 35$). As shown in Fig. 6d, 25 out of 35 ascites contained sE-cad and sedimented in the middle of the gradient equivalent to the density of exosomes, confirming that the cleavage of E-cadherin in released vesicles that we showed in cell cultures might be found in situ in ovarian carcinomas. This finding would also suggest that exosomes may be a factor responsible for the release of sE-cad in vivo. Expression of sE-cad-positive exosomes was defined as high ($+++$) (>20 μg mL$^{-1}$), medium ($++$) (10–20 μg mL$^{-1}$), low ($+$) (<10 μg mL$^{-1}$), and none ($-$) (0 μg mL$^{-1}$) levels. Our analyses revealed high levels of sE-cad-positive exosomes (2.62–27.10 μg mL$^{-1}$) in the ascitic fluids of ovarian carcinomas (Supplementary Table 1). There was no strong association between the expression levels of sE-cad-positive exosomes and tumor histologic subtypes (Supplementary Table 1), which were consistent with those of the cell lines (Supplementary Fig. 9). In contrast, peritoneal fluids of non-cancerous (benign) patients exhibited no or only weak (3.92–8.13 μg mL$^{-1}$) expression of sE-cad-positive exosomes (Supplementary Table 2). In accord with the ovarian cancer cell lines data, patient ascites-derived exosome was a potent inducer of endothelial tube formation (Fig. 6f). The enhanced number and length of capillary tubes formed by endothelial cells were blocked in the presence of the mAb to E-cadherin (HECD-1), but not to nonimmune IgG (Fig. 6f). Moreover, treatment with patients'-derived exosomes substantially augmented ascites formation, tumor dissemination, and neovascularization as compared with the control group, and that the effects were abrogated in the presence of HECD-1 (Fig. 6g, h), further supporting the rationale of therapeutic targeting of sE-cad.

## Discussion

It is well recognized that tumor cells can modify their microenvironment using proteolytes[27]. The critical question is whether the generation of sE-cad is merely a byproduct of tumor cells or offers advantages to the tumor cells. In the present study, we show for the first time sE-cad is a potent inducer of angiogenesis. Notably, we also provide evidence for an important role of extracellular vesicles in the constitutive release of sE-cad from tumor cells both in vitro and in vivo. As exosomes isolated from ovarian cancer patients ascites are also functionally active, our findings provided new insight into the clinical correlation between levels of sE-cad in ascites and the advanced stage of ovarian cancer.

We demonstrated that sE-cad stimulates (a) endothelial cell migration, (b) permeability, (c) endothelial cell tube formation, and (d) neovascularization in Matrigel implants. However, it does not act as an endothelial cell mitogen, which has been demonstrated for cell adhesion molecules such as ICAM-1, E-selectin, and VCAM-1[28,29]. These findings can be interpreted in light of their potent migration potential, which is most closely associated with the angiogenic function, may be responsible for its angiogenic activity[30].

---

**Fig. 2** Intracellular cleavage of sE-cad and the release of sE-cad-positive exosomes. **a** Western blot analysis of E-cadherin in total cell lysate and conditioned medium (CM) of Caov-3 and OV-90. **b** Western blot analysis of the subcellular localization of E-cadherin in Caov-3 and OV-90 by sucrose density gradient fractionation. Ves vesicles, TGN trans-Golgi network, ER endoplasmic reticulum, PM plasma membrane. Upper: Representative western blot images. Lower: Densitometry of FL-E-cad. **c** Western blot analysis of the protein composition of exosomes isolated from Caov-3 and OV-90. **d** Sucrose density gradient fractionation of exosomes isolated from Caov-3 and OV-90. Upper: Densities (g cm$^{-3}$) of each sucrose fraction as determined by refractometry. Lower: Representative images of western blot. **e** Immunoelectron microscopic view of the sE-cad localization on the exosomal surface of Caov-3 and OV-90 cells at a magnification of 52,000×. Bar, 100 nm. Localization of CD63 as a positive control was visualized. No staining was found on exosomes incubated with an isotype-matched control IgG. **f** The number of particles per μg exosomes and median diameter were determined by nanoparticle tracking analysis. $n = 3$ per group, all experiments were repeated three times. Error bar indicates SD of the mean

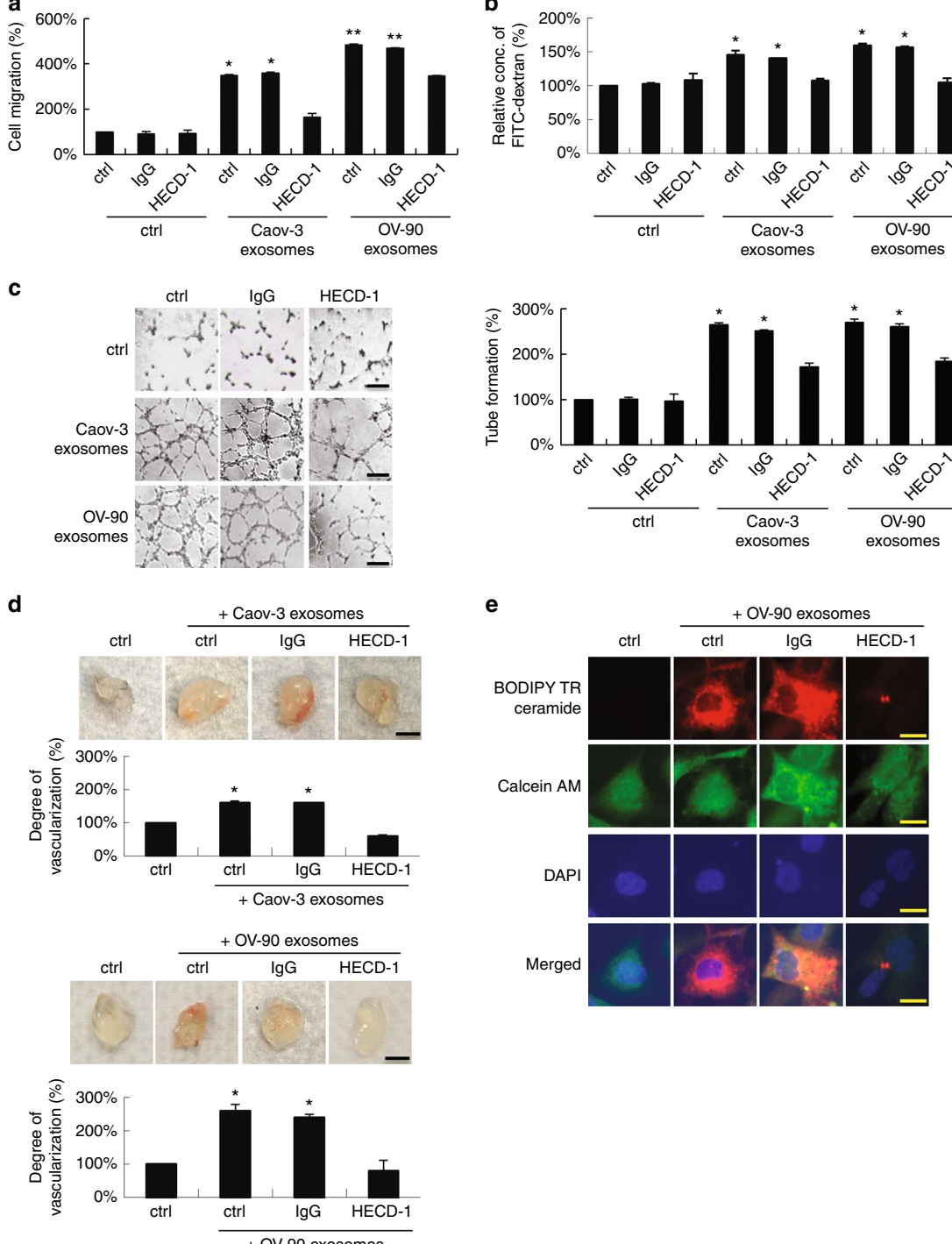

**Fig. 3** sE-cad-positive exosomes promote angiogenesis in vitro and in vivo. **a** Cell migration assay of HUVEC treated with exosomes (25 μg mL$^{-1}$) in the presence or absence of E-cadherin neutralizing antibodies, HECD-1 (100 μg mL$^{-1}$). **b** Permeability analysis of HUVEC measured by the percentage change of FITC–dextran flux (excitation 485 nm, emission 535 nm) treated with exosomes (25 μg mL$^{-1}$) in the presence or absence of E-cadherin neutralizing antibodies, HECD-1 (100 μg mL$^{-1}$). **c** Tube formation assay of HUVEC treated with exosomes (25 μg mL$^{-1}$) in the presence or absence of E-cadherin neutralizing antibodies, HECD-1 (100 μg mL$^{-1}$). Left: Representative images of HUVEC tube formation assay. Right: Quantification of the percentage change of the number of branching points. Bar, 100 μm. **d** In vivo Matrigel plug implant model using C57/BL6 mice subcutaneously injected with Matrigel containing exosomes (25 μg mL$^{-1}$) in the presence or absence of E-cadherin neutralizing antibodies, HECD-1 (100 μg mL$^{-1}$). In vivo neovascularization is measured by the Drabkin's reagent kit 7 days after injection. Upper: Representative images of excised Matrigel plug. Lower: Quantification of the percentage change of hemoglobin content. Bar, 5 mm. **e** HUVECs were treated with exosomes (25 μg mL$^{-1}$) in the presence or absence of E-cadherin neutralizing antibodies, HECD-1 (100 μg mL$^{-1}$). Red: BODIPY TR ceramide; Green: Calcein AM; Blue: DAPI. Bar, 5 μm. For in vivo Matrigel plug model, $n = 6$ per group, and were conducted twice. For the other assays, $n = 3$ per group, all experiments were repeated three times. Error bar indicates SD of the mean. *$P < 0.05$, **$P < 0.01$ versus untreated control using one-way analysis of variance followed by Tukey's least significant difference post hoc test

The source of sE-cad in ovarian cancer is unclear: it can be generated by ectodomain shedding, and it is likely to involve more than one source. Our results suggest that sE-cad may be arising from the tumor itself. Similar observations were reported previously[9]. Our findings are the first to demonstrate sE-cad cleavage from exosomes, adding E-cadherin as a new member of exosomal protein and suggesting that sE-cad may play a role in the crosstalk between the tumor and the tumor microenvironment that supports disease progression. Of interest, intact other than the cleaved form of E-cadherin was found on exosomes. Indeed, intact E-cadherin may be more advantageous compared with the soluble protein as the exosomal membrane, which is rich in cholesterol and sphingomyelin, helps to maintain a stable conformation of the cell surface protein for a half-life that is

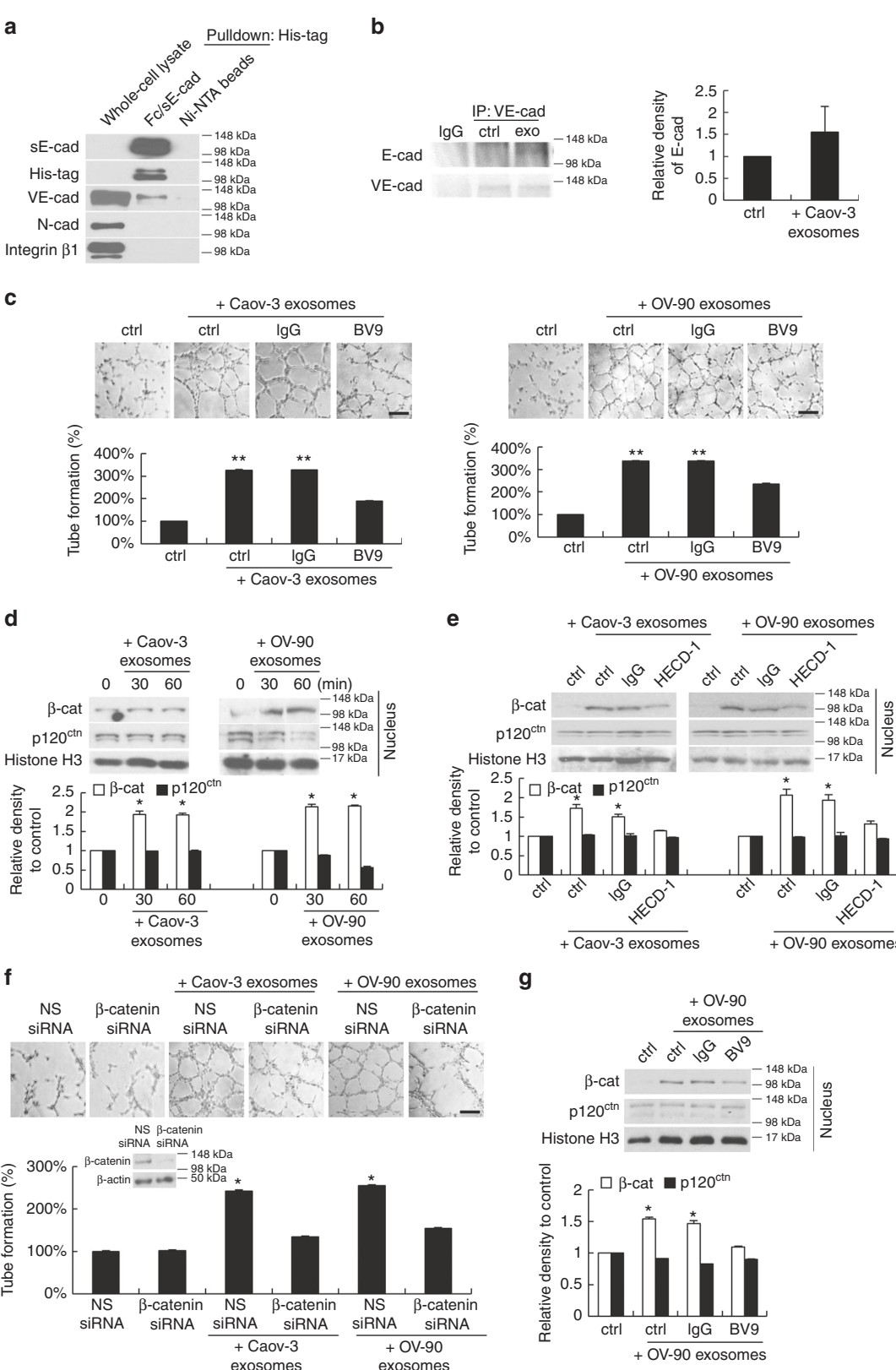

much longer (in some cases it can be as long as 90 days) compared to the soluble protein (from minutes to hours), thus enabling it to travel over long distance and displays significantly more effective cancer phenomena[31,32].

sE-cad-positive vesicles were also observed in the ascites of ovarian carcinoma patients. Several observations could be made: (a) ascites-derived exosomes had similar density features as exosomes secreted by human ovarian cancer lines; (b) sE-cad-positive exosomes in the ascitic fluid induce angiogenesis, and (c) selective targeting of sE-cad in vivo reduces ascites formation. Indeed, tumor-derived exosomes are commonly found in the ascites of ovarian cancer patients. Although mechanisms have not been fully elucidated, the cell surface molecules, EpCAM and CD24[33], have been shown to exert gelatinolytic activities that may contribute to tumor spreading. Exosomes isolated from a cisplatin-resistant human ovarian cancer cell line 2008/C13*5.25 are able to export cisplatin[34], and exosomes derived from tumor cells and dendritic cells are linked to anti-tumor immune responses[35,36]. There are also emerging evidences suggesting that tumor cell-derived exosomes could modify and initiate a pre-metastatic niche[37,38].

In search of mechanisms underlying sE-cad regulation of angiogenesis, we found that sE-cad mediates heterophilic trans-connection with VE-cadherin. This provides a novel and attractive mechanism whereby endothelial cells, which lack E-cadherin, can lead to angiogenic signaling. Although previously thought to interact in an exclusively homophilic manner, there is now intriguing evidence that different cadherins can be associated in heterophilic binding[39]. These heterophilic interactions may serve as a new paradigm of cell–cell communication[39]. Heterodimer cadherin complexes have been reported to form between N- and R-cadherin using L cell transfections[40], N- and E-cadherin in endoderm-derived cells[41], and L1- and E-cadherin in intestinal epithelium[42]. Our finding is the first to report heterodimeric E-/VE-cadherin complex and also define a novel heterotypic connection between tumor cell and its microenvironment. Our data (Supplementary Fig. 10) and previous studies showed that β-catenin is present in exosomes[43]. Thus, although we cannot exclude the possibility that β-catenin originates from the exosomes in mediating the angiogenic response, our findings delineate a novel mechanism that links β-catenin and NFκB. Whereas crossregulation between β-catenin and NFκB has been generally reported[23–25], we found that they were two parallel pathways. Sequential activation of β-catenin and NFκB has not been reported, but a possible interplay of β-catenin and other signaling pathway during embryonic development is recently suggested[44]. The sequential activation is known to allow continuous input from the signaling pathway for sustaining transcriptional and subsequent cellular response. The combined operation of different mechanisms suggests that the interaction of sE-cad/VE-cadherin is a crucial event and emphasizes the important role of sE-cad/VE-cadherin that has in contributing to the angiogenic process.

VEGF has been considered as one of the most promising therapeutic anti-angiogenic targets[45–50]. Bevacizumab, a recombinant humanized monoclonal antibody that targets VEGF, has been used in the clinic and shown significant progression-free survival benefit as a single agent or in combination with front-line chemotherapy carboplatin and paclitaxel in advanced or recurrent ovarian cancer patients[51,52]. However, although bevacizumab is effective, it and other VEGF inhibitors also interfere with the normal VEGF pathway, leading to numerous serious adverse side effects[53]. The acquisition of resistance to therapy also necessitates new strategies targeting angiogenesis. We found that sE-cad possessed similar VEGF efficiency in angiogenesis, but in a VEGF-independent manner. VEGF-independent angiogenesis pathways were reported in ovarian and other cancers[54,55], suggesting the existence of alternative pro-angiogenic factors and signaling molecules. Importantly, targeting sE-cad might have an advantage over VEGF, as it is less sensitive to some resistance mechanisms developed by cancer cells[56]. Moreover, targeting sE-cad could provide the best approach not only on primary tumor angiogenesis, but also at disrupting the vascular barrier that contributes to ascites formation, the most distressing complication of ovarian cancer[57]. The specific expression in tumor tissues may be hypothesized to limit adverse off-target toxicities. Recent evidence suggests that mAb DECMA-1, which preferentially reacts with the shed ectodomain sE-cad fragment as compared to intact E-cadherin on normal cells, can be exploited for tumor-targeted therapy[58]. However, sE-cad has only been characterized in anti-cancer cell proliferation effects. Here, we showed for the first time that sE-cad may be involved in other aspects of tumor progression, such as angiogenesis. These results reveal a new direction for understanding the oncogenic roles of sE-cad in tumor progression. High serum sE-cad level has also been detected in patients with breast, gastric, and colon cancer, and its expression positively correlates with the aggressive behavior[59]. We found that sE-cad containing exosomes were present and able to induce angiogenesis in MCF-7 breast and HCT116 colon cancer cells (Supplementary Fig. 11). Moreover, sE-cad-positive exosomes can also be found in the ascitic fluids of patients with colon, breast, liver, endometrial, and stomach cancer (Supplementary Table 2), suggesting that the angiogenic function of sE-cad that we propose may have broader implications for other tumor types as well.

**Fig. 4** Heterodimerization of sE-cad with VE-cadherin and the activation of VE-cadherin/β-catenin signaling cascade. **a** His-tag pull down assay using sE-cad/Fc (10 μg mL$^{-1}$) on Ni-NTA beads and HUVEC lysate (1 mg). **b** Immunoprecipitation assay using anti-VE-cadherin in HUVEC lysates (1 mg) untreated or treated with exosomes (25 μg mL$^{-1}$). Left: Representative western blot images. Right: Densitometry of E-cadherin. **c** Tube formation assay of HUVEC treated with exosomes (25 μg mL$^{-1}$) in the presence or absence of VE-cadherin neutralizing antibodies, BV9 (10 μg mL$^{-1}$). Upper: Representative images of HUVEC tube formation assay. Lower: Quantification of the percentage change of the number of branching points. Bar, 100 μm. **d** Subcellular fractionation of HUVEC treated with exosomes (25 μg mL$^{-1}$) for 0, 30, or 60 min. Upper: Representative images of nuclear fractions. Lower: Densitometry of β-catenin (β-cat) and p120 catenin (p120$^{ctn}$). **e** Subcellular fractionation of HUVEC treated with exosomes (25 μg mL$^{-1}$) in the presence of E-cadherin neutralizing antibodies, HECD-1 (100 μg mL$^{-1}$) for 60 min. Upper: Representative images of nuclear fractions. Lower: Densitometry of β-catenin (β-cat) and p120 catenin (p120$^{ctn}$). **f** Tube formation assay of HUVEC transiently transfected with non-specific (NS) siRNA or β-catenin siRNA (20 nM) for 24 h, treated with exosomes (25 μg mL$^{-1}$). Upper: Representative images of HUVEC tube formation assay. Lower: Quantification of the percentage change of the number of branching points. Representative western blot images of β-catenin knockdown in HUVEC were also included. Bar, 100 μm. **g** Subcellular fractionation of HUVEC treated with exosomes (25 μg mL$^{-1}$) in the presence of VE-cadherin neutralizing antibodies, BV9 (10 μg mL$^{-1}$) for 60 min. Left: Representative images of nuclear fractions. Right: Densitometry of β-catenin (β-cat) and p120 catenin (p120$^{ctn}$). $n = 3$ per group, all experiments were repeated three times. Error bar indicates SD of the mean. *$P < 0.05$ versus untreated control using one-way analysis of variance followed by Tukey's least significant difference post hoc test

## Methods

**Cell culture and patient samples**. The OVCAR-3, Caov-3, and OV-90 human ovarian cancer cell lines, the MCF-7 human breast cancer cell line, and the HCT116 prostate cancer cell line were obtained from ATCC and maintained in standard conditions as described[60]. The MCAS and OVK18 (gifts from Dr. L. Cheung, School of Biomedical Sciences, University of Hong Kong) were maintained in MEM containing 20% FBS and RPMI containing 5% FBS, respectively. TOV21G and TOV112D (gifts from Dr. D. Chan, Department of Obstetrics and Gynecology, University of Hong Kong) were maintained in M199:MCDB105 containing 15% FBS. HUVECs and HMVECs were purchased from Clonetics (San Diego, CA) and cultured in F12K supplemented with FBS (10%), endothelial cells

growth supplement (20 µg mL$^{-1}$), heparin (90 U mL$^{-1}$), penicillin (100 U mL$^{-1}$), and streptomycin (100 µg mL$^{-1}$). The third to eighth passages of HUVECs or HMVECs were used in these studies to ensure genetic stability of the culture. All cells were kept in a humidified incubator with 5% $CO_2$ at 37 °C. Primary tumor ascitic samples were obtained from 35 ovarian cancer patients and 15 patients with various other cancer types. Peritoneal fluids were obtained from six patients operated from conditions other than cancer. Normal human OSE were derived from surface scrapings of normal ovaries from women with nonmalignant gynecological diseases and FTE were obtained from patients undergoing bilateral salpingo-oophorectomies. OSE and FTE were cultured in M199: MCDB105 supplemented with 15% FBS. The use of these specimens was approved

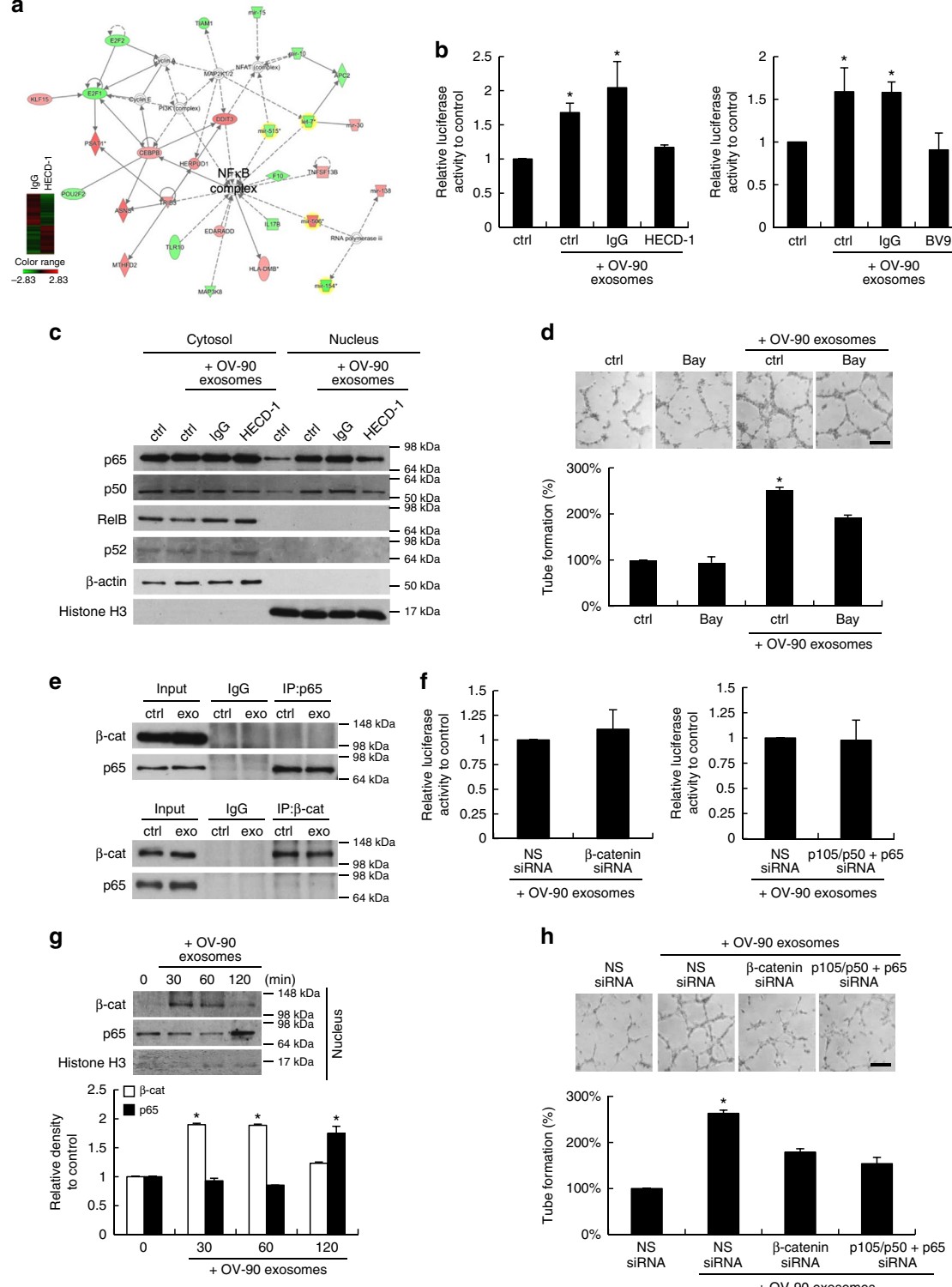

by the Institutional Ethical Review Board for Research on the use of human subjects at the University of Hong Kong.

**Conditioned medium and immunodepletion**. Cells were starved in serum-free medium for 24 h after washing with PBS twice. Conditioned medium were collected, centrifuged at 1000$g$ for 5 min and filtered through a 0.2 µm filter. Immunodepletion of sE-cad was performed by incubating the conditioned medium with 100 µg mL$^{-1}$ HECD-1 antibodies (Invitrogen, Carlsbad, CA) overnight at 4 °C with shaking. The antibodies–protein complex was removed by incubating with protein A/G PLUS agarose beads (Santa Cruz Biotechnology, Santa Cruz, CA).

**Cytotoxicity assay**. Cell survival was determined using the MTT assay according to the manufacturer's instructions (Sigma) and as previously described[61]. Briefly, cells were incubated with MTT solution for 4 h at 37 °C, medium was then aspirated and dissolved in 250 µL DMSO. The colorimetric analysis was measured at 570 nm with a microplate reader (Bio-Rad).

**Migration assay**. HUVECs or HMVECs ($3 \times 10^4$) were seeded onto 8 µm pore Transwell inserts (Millipore) in a 24-well plate. Cell migration was examined 24 h after conditioned media or exosomes and/or HECD-1 treatment and stained with 0.5% crystal violet.

**In vitro permeability assay**. HUVECs or HMVECs ($3 \times 10^4$) were seeded onto Transwells (0.4 µm pore; Millipore). After the endothelial monolayer formed, conditioned media or exosomes and/or HECD-1 was treated. FITC–dextran (1 µg µL$^{-1}$; Sigma, St. Louis, MO) was added to the upper chamber and the presence of FITC–dextran in the lower chamber after 20 min was determined by fluorescent measurement (Victor X3—PerkinElmer).

**Tube formation assay**. The tube formation assay was performed as described previously[62]. HUVECs or HMVECs ($1 \times 10^5$) were seeded in a growth factor-reduced Matrigel-coated 24-well plate. Cells were untreated or treated with conditioned media or exosomes in the absence or presence of E-cadherin (HECD-1, Invitrogen, Carlsbad, CA), VE-cadherin (clone BV9, Abcam, Cambridge, MA) or VEGF-neutralizing antibodies (clone 26503, R&D System, Minneapolis, MN) or NFκB inhibitor (Bay11-7082, Sigma, St. Louis, MO) for 6 h at 37 °C. Images were captured under phase contrast microscopy (×10) using a CCD camera. Twelve microscopic fields were randomly selected for each well, and the number of branch points of the tubes per field was counted.

**In vivo Matrigel plug assay**. Growth factor-reduced Matrigel (BD Biosciences) mixed with Fc/sE-cad or sE-cad-positive exosomes in the absence or presence of HECD-1 blocking antibodies were injected subcutaneously into 6-week-old C57/BL6 mice. On day 7, the mice were sacrificed and Matrigels were excised to assess the angiogenic response. The formation of neovessels was indirectly determined by measuring the hemoglobin content using the Drabkin's reagent kit (Sigma, St. Louis, MO) according to the manufacturer's protocol. Six mice were used as a group and the experiment was done in duplicate.

**Subcellular fractionation**. Subcellular fractionation was performed as described previously[63]. Briefly, cells were lysed in lysis buffer containing 10 mM HEPES, 10 mM NaCl, 1 mM KH$_2$PO$_4$, 5 mM NaHCO$_3$, 5 mM EDTA, 1 mM CaCl$_2$, 0.5 mM MgCl$_2$, and supplemented with protease inhibitors as described previously. After incubating on ice for 5 min, cells were homogenized and tonicity was restored by adding 2.5 M sucrose. Homogenate was centrifuged at 6300$g$ for 5 min at 4 °C. The resulting pellet was resuspended in TSE buffer containing 10 mM Tris, 300 mM

sucrose, 1 mM EDTA, 0.1% IGEPAL-CA 630 (v/v), pH 7.5, and further lysed with a homogenizer. This homogenate was subjected to centrifugation at 4000$g$ for 5 min, the resulting pellet was repeatedly washed in TSE buffer. The final pellet, which is the nucleus, was resuspended in TSE buffer.

**Sucrose density gradient fractionation**. Cells were homogenized in buffer containing 10 mM Tris–HCl, 1 mM EDTA, pH 7.4. Homogenate or pelleted vesicles were loaded onto a step gradient comprising layers of 2, 1.3, 1.16, 0.8, 0.5, and 0.25 M sucrose as described by Gutwein et al.[64]. The gradients were centrifuged at 100,000$g$ for 2.5 h. Twelve fractions were collected from the top of the gradient. Samples were recovered by chloroform/methanol precipitation (1:4, v/v) and analyzed by SDS PAGE and western blotting as described below. The amount of exosomal protein was quantified by DC Protein Assay (Bio-Rad). The particle size and concentration were estimated with Nanoparticle Tracking Analysis (ZetaView).

**Exosomes isolation and characterization**. Exosomes were isolated as described previously[65]. Briefly, conditioned media or patients' ascitic or peritoneal fluids were subjected to differential centrifugation (300$g$ for 10 min, 2000$g$ for 10 min, and 10,000$g$ for 30 min). The resulting supernatant was filtered through a 0.2 µm filter. Filtered supernatant was ultracentrifuged at 100,000$g$ for 70 min. Pellet was washed with PBS and further centrifuged at 100,000$g$ for 70 min. The resulting pellet consisting of exosomes was resuspended in PBS and stored at −80 °C for further analysis. For patient-derived exosomes, sE-cad level "high (+++), medium (++), low (+), and none (−)" refer to >20 µg mL$^{-1}$, 10–20 µg mL$^{-1}$, <10 µg mL$^{-1}$, and 0 µg mL$^{-1}$, respectively.

**Electron microscopy**. Electron microscope examination of exosomes was carried out by depositing fixed exosomes onto a Formvar-carbon-coated electron microscope grid for 20 min. For immunoelectron labeling, the grid was incubated with HECD-1 or CD63 antibodies, followed by mouse secondary antibodies conjugated with 10 nm gold particles. The grid was washed in PBS and fixed in 2% glutaraldehyde (Electron Microscopy Sciences) for 5 min. The grid was then washed with water and contrasted in uranyl oxalate (Electron Microscopy Sciences) for 5 min. Samples were then embedded in uranyl acetate and methyl cellulose for 10 min on ice. Air-dried grids were visualized using a Philips CM 100.

**Exosomes internalization**. Exosomes were labeled with 10 µM BODIPY TR ceramide (Invitrogen) according to the manufacturer's instruction. Excess dyes were removed with an exosomes spin column (MW 3000) (Invitrogen). In addition, to control for incomplete dye removal, we columned purified dye in the absence of sE-cad-positive exosomes. HUVECs ($5 \times 10^4$) were labeled with 2.5 µg mL$^{-1}$ Calcein AM (Invitrogen). Labeled exosomes (25 µg mL$^{-1}$) were then added to HUVECs for 24 h. Cells were washed with PBS twice and fixed with 4% paraformaldehyde, the nuclei were counter-stained with 4′,6-diamidino-2-phenylindole and mounted with Vectashield (Vector Laboratories, Burlingame, CA). The localization of fluorescent signals was determined by fluorescent microscopy (Nikon, Tokyo, Japan).

**Western blot analysis**. Cells were lysed with buffer containing 260 mM Tris–HCl, pH 6.8, 0.8% SDS (w/v), and 40% glycerol, supplemented with protease inhibitors: 1 µg mL$^{-1}$ aprotinin, 1 µg mL$^{-1}$ leupeptin, 1 µg mL$^{-1}$ pepstatin, and 1 mM phenylmethyl sulfonyl. Equal amounts of protein (20 µg) were resolved by SDS-PAGE for blotting using anti-HECD-1 (13-1700, 1:1000) (Invitrogen); anti-E-cadherin (610182, 1:1000), anti-Hsp70 (610607, 1:1000), anti-Tsg101 (612696, 1:1000), anti-nucleoporin p62 (610497, 1:1000), anti-cytochrome c (556433, 1:1000), anti-GRP78 (610978, 1:1000), anti-p120 catenin (610134, 1:1000), and anti-β-catenin

**Fig. 5** sE-cad-positive exosomes activate the NFκB pathway. **a** Top network identified by IPA. Gene signatures of HUVEC incubated with IgG pre-treated exosomes and HECD-1 pre-treated exosomes. **b** NFκB reporter assay of HUVEC treated with exosomes (25 µg mL$^{-1}$) in the presence or absence of E-cadherin neutralizing antibodies, HECD-1 (100 µg mL$^{-1}$) or VE-cadherin neutralizing antibodies, BV9 (10 µg mL$^{-1}$). **c** Western blot analysis of the subcellular localization of NFκB subunits in HUVEC treated with exosomes (25 µg mL$^{-1}$) in the presence or absence of E-cadherin neutralizing antibodies, HECD-1 (100 µg mL$^{-1}$). **d** Tube formation assay of HUVEC treated with exosomes (25 µg mL$^{-1}$) in the presence or absence of Bay11-7082 (Bay) (10 µM). Upper: Representative images of HUVEC tube formation assay. Lower: Quantification of the percentage change of the number of branching points. Bar, 100 µm. **e** Coimmunoprecipitation of NFκB subunits and β-catenin (β-cat) in HUVEC with or without exosomes treatment (25 µg mL$^{-1}$). **f** NFκB reporter assay of HUVEC transiently transfected with non-specific (NS) siRNA or β-catenin (β-cat) siRNA (20 nM) for 24 h, treated with exosomes (25 µg mL$^{-1}$). TCF reporter assay of HUVEC transiently transfected with NS siRNA or p105/p50 and p65 siRNA (20 nM) for 24 h, treated with exosomes (25 µg mL$^{-1}$). **g** Western blot analysis of the β-catenin (β-cat) and NFκB subunits in HUVEC treated with exosomes (25 µg mL$^{-1}$) for the indicated period of time. Upper: Representative images of nuclear fractions. Lower: Densitometry of β-catenin (β-cat) and p65. **h** Tube formation assay of HUVEC transiently transfected with non-specific (NS) siRNA, β-catenin siRNA, or p105/p50 siRNA and p65 siRNA (20 nM) for 24 h with or without exosomes treatment (25 µg mL$^{-1}$). Upper: Representative images of HUVEC tube formation assay. Lower: Quantification of the percentage change of the number of branching points. Bar, 100 µm. $n = 3$ per group, all experiments were repeated three times. Error bar indicates SD of the mean. *$P < 0.05$ versus untreated control using one-way analysis of variance followed by Tukey's least significant difference post hoc test

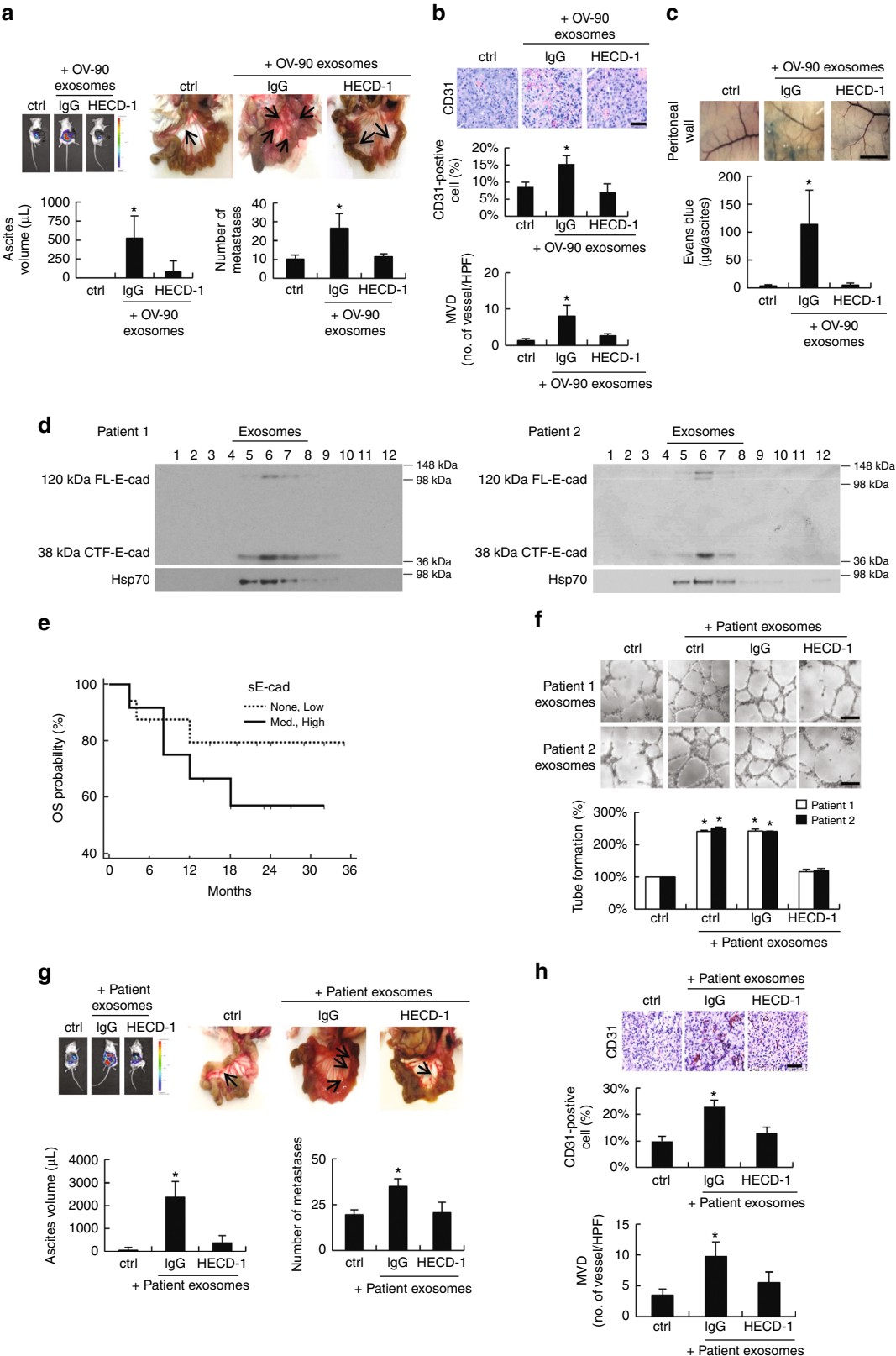

(610153, 1:1000) (BD Biosciences, San Diego, CA); and anti-CD63 (ab8219, 1:1000) (Abcam, Cambridge, MA); anti-histone H3 (9715, 1:1000), anti-VE-cadherin (2500, 1:1000), anti-NFκB p65 (6956, 1:1000), anti-NFκB p105/p50 (12540, 1:1000), anti-NFκB p100/52 (4882, 1:1000), and anti-RelB (4922, 1:1000) (Cell signaling, Austin, TX); anti-integrin β1 (MAB1965, 1:1000) (Chemicon, Billerica, MA); anti-His-tag (70796-3, 1:1000) (Novagen, Darmstadt, Germany); anti-N-cadherin (C3865, 1:1000), anti-GM130 (G7295, 1:1000), and anti-β-actin

(A5316, 1:5000) (Sigma, St. Louis, MO). The membrane was washed and incubated with appropriate secondary antibodies (1:3000) (Bio-Rad) for 1 h. Immunor-eactivities were visualized using the enhanced chemiluminescence system (Perkin Elmer, Waltham, MA). β-actin was probed to ensure equal protein loading. For analyzing conditioned medium, equal loading was determined by visualizing total protein level with Coomassie blue staining (Bio-Rad). Scans of the full blots are shown in Supplementary Figs. 12–18.

**Fig. 6** The angiogenic role of sE-cad-positive exosomes is clinically relevant. **a** Intraperitoneal dissemination assay using OV-90-derived exosomes. Upper left: Representative images of bioluminescence record. Upper right: metastasis in the peritoneal cavity, arrows indicate tumor. Lower: Quantification of the number of metastases and aspirated ascitic fluid. **b** Immunohistochemical staining of CD31. Upper: Representative images of immunohistochemical staining. Middle: Quantification of CD31-positive cells. Lower: Microvessel density (MVD) was expressed as number of vessels per high-power field (HPF). Bar, 50 μm. **c** NOD/SCID mice bearing HEYA8 tumors were administered with Evans blue. Upper: Representative image of Evans blue leakage in the abdominal wall. Lower: Ascites were aspirated and Evans blue content was quantified. Bar, 5 mm. **d** Sucrose density gradient fractionation of exosomes isolated from ovarian cancer patients' ascites. The presence of E-cadherin is determined by western blotting. **e** Kaplan–Meier overall survival (OS) analysis of clinical samples of none–low or med–high sE-cad-positive exosomes expression, significance were analyzed using the log-rank test. **f** Tube formation assay of HUVEC treated with patients-derived exosomes (25 μg mL$^{-1}$) in the presence or absence of E-cadherin neutralizing antibodies, HECD-1 (100 μg mL$^{-1}$). Upper: Representative images of HUVEC tube formation assay. Lower: Quantification of the percentage change of the number of branching points. **g** Intraperitoneal dissemination assay using patient-derived exosomes. Upper left: Representative images of bioluminescence record. Upper right: metastasis in the peritoneal cavity, arrows indicate tumor. Lower: Quantification of the number of metastases and aspirated ascitic fluid. **h** Immunohistochemical staining of CD31. Upper: Representative images of immunohistochemical staining. Middle: Quantification of CD31-positive cells. Lower: MVD was expressed as number of vessels per HPF. Bar, 50 μm. $n = 3$ per group; and the experiment was conducted at least twice. Error bar indicates SD of the mean. *$P < 0.05$ versus untreated control using one-way analysis of variance followed by Tukey's least significant difference post hoc test

**Pull down assay**. HUVEC was lysed in lysis buffer for native conditions (50 mM NaH$_2$PO$_4$, 300 mM NaCl, 10 mM imidazole, 1% NP-40, pH 8.0) supplemented with protease inhibitors. Cells were homogenized and centrifuged at 14,000 rpm for 5 min. Supernatant (1 mg) was then incubated with 10 μg mL$^{-1}$ sE-cad/Fc for 6 h on a shaker at 4 °C to facilitate protein–protein interaction. The suspension was further incubated with Ni-NTA for 1.5 h. Protein–beads complex was collected by centrifugation at 1000$g$ for 5 min. Beads were washed with washing buffer (50 mM NaH$_2$PO$_4$, 300 mM NaCl, 20 mM imidazole, pH 8.0) for three times. Protein was eluted by incubating with elution buffer (50 mM NaH$_2$PO$_4$, 300 mM NaCl, 500 mM imidazole, pH 8.0) for 15 min. The elution product was subjected to western blot analysis.

**Immunoprecipitation**. Nuclear extracts were lysed in immunoprecipitation buffer (2 mM EDTA, 50 mM Tris, pH 7.5, 150 mM NaCl, and 1% NP-40) supplemented with protease inhibitors as described previously[66]. Equal amounts of protein (1 mg) were precleared with protein A/G PLUS agarose beads (Santa Cruz Biotechnology) for 2 h. Supernatant was then incubated with anti-p65 antibodies or mouse IgG at 4 °C overnight. Protein–antibodies complexes were collected by incubating with protein A/G PLUS agarose beads (Santa Cruz Biotechnology) for 4 h. Non-specific binding was removed by washing with immunoprecipitation buffer. Final products were eluted with sample buffer (125 mM Tris, pH 6.8, 20% glycerol, 1.6% 2-mercaptoethanol, and 1% SDS) and subjected to western blot analysis.

**siRNA transfection**. β-catenin-specific siRNA (J-003482-09), p105/p50-specific siRNA (D-003520-01), and p65-specific siRNA (D-003533-03) were purchased from Dharmacon (Lafayette, CO). A non-targeting siRNA was used as control (D-001210-01). Cells ($2.5 \times 10^5$ per well) were seeded in 6-well plates and transfected overnight with 20 nM siRNA using siLentFect (Bio-Rad) according to the manufacturer's protocol.

**Microarray analysis**. Total RNA was extracted with TRIzol reagent (Invitrogen) according to the manufacturer's protocols. Briefly, the RNA was quantified with NanoDrop1000 and its quality was checked with an Agilent 2100 Bioanalyzer. 500 ng of total RNA were amplified and labeled to generate biotin-labeled cDNA using the GeneChip WT PLUS Reagent kit and then hybridized onto the GeneChip Human Gene 2.0 ST Array using the GeneChip Hybridization kit (Affymetrix), according to the manufacturer's protocols. After each array was washed and stained using the Fluidics Station 450, CEL files were generated by scanning with the GeneChip Scanner 7G with Command Console® (AGCC) software. The array quality was assessed with the Expression Console software (version 1.4). The data were normalized with the robust multiarray averaging (RMA) method using the Partek Genomics Suite 6.6. An expression ratio of 1.5 was chosen to define genes that are differentially expressed. IPA (Ingenuity Systems) was used to generate networks and assess statistically-relevant functions and pathways.

**Luciferase reporter assay**. Briefly, cells ($2.5 \times 10^5$ per well) were seeded in 6-well plates and transfected overnight with 1 μg of NFκB, TOPFLASH, or FOPFLASH reporter construct (Clontech, Palo Alto, CA) and cotransfected with 0.5 μg of β-galactosidase expression plasmid (pSV-β-gal; Promega, Madison, WI), using Lipofectamine 2000 (Life Technologies). 24 h after transfection, cells were treated with exosomes with or without HECD-1 blocking antibodies for 6 h, and then harvested with reporter lysis buffer (Promega). Luciferase activity was assayed using the luciferase assay kit according to the manufacturer's protocol (Promega). β-galactosidase activity was used to normalize the transfection.

**Orthotopic in vivo ovarian cancer model**. All mouse studies were performed in accordance with protocols approved by the University of Hong Kong Animal Care

and Use Committee. Female NOD/SCID mice were purchased from the Charles River Laboratories and maintained as previously described[60]. HEYA8 ($1 \times 10^6$) were transduced with lentivirus encoding luciferase and used for intraperitoneal injection into NOD/SCID mice ($n = 3$ mice per group), and the experiment was conducted twice. OV-90 (25 μg per dose) or ovarian cancer patient-derived exosomes (15 μg per dose) pre-treated with IgG or HECD-1 were given intraperitoneally twice a week over a period of 21 days. The mice were imaged once weekly for bioluminescence signal using the Xenogen IVIS system. At the time of sacrifice, the volume of ascites was determined with a pipette and the number of all visible (>0.1 cm) metastatic nodules in the peritoneal cavity was counted. Tissue specimens were fixed with formalin. The number of blood vessels was stained for CD31 (1:50 dilution, Abcam, Cambridge, MA) and counted in five random fields at 200× magnification. All staining was quantified by two investigators in a blinded fashion. To assess the extent of vascularization, counting was performed in the three highest MVD areas at high power (200×). To measure vascular leakiness, Evans blue (80 mg kg$^{-1}$) (Sigma, St. Louis, MO) were injected through the tail vein and circulated for 40 min. Evans blue dye was extracted from the ascitic fluid and its content was quantified spectrophotometrically at 595 nm.

**Statistical analysis**. Data were expressed as the mean ± SD of at least three independent experiments. Differences between control and treatment were compared for significance using the Student's test. Statistical analyses of three or more groups were compared using one-way analysis of variance (ANOVA) followed by Tukey's least significant difference post hoc test (GraphPad, San Diego, CA). $P < 0.05$ was considered statistically significant. Survival estimates were computed using the Kaplan–Meier plot, and comparisons between none–low and med–high groups were analyzed using the log-rank test.

**Data availability**. The GEO accession number for the microarray data reported in this paper is GSE95174. All other remaining data are available within the Article and Supplementary Files, or available from the authors upon request.

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

## Acknowledgments

This work is supported by Hong Kong Research Grant Council grant (781013). A.S.T.W. is a recipient of the Croucher Senior Research Fellowship.

## Author contributions

M.K.S.T. conducted the experiments, and analyzed and interpreted the results. R.L.H., H.C.L., P.P.I., A.N.Y.C., K.Y.T., and H.Y.S.N. secured surgical patient samples and conducted pathological examination of patients' samples and tests for association between sE-cad and clinicopathologic variables. P.Y.K.Y. helped in in vivo angiogenesis assays and provided technical advice. A.S.T.W. designed and supervised the project. M.K.S.T. and A.S.T.W. wrote the draft manuscript and all authors reviewed the manuscript.

## Additional information

**Competing interests:** The authors declare no competing interests.

