## [Peer Review File · Nature Communications]

Reviewers' comments:

Reviewer #1 (Remarks to the Author):

In the present manuscript, the authors report that sE-cadherin stimulates angiogenesis and is released by ovarian cancer cells packaged in exosomes. The authors speculate that exosome-derived sE-cad facilitates ovarian tumor growth through stimulation of angiogenesis. Therefore, targeting sE-cad could improve the outcomes of antiangiogenic therapy with VEGF inhibitors in ovarian cancer.

The hypothesis is interesting and the authors provide some evidence that sE-cad affects at least some aspects of angiogenesis in vitro assays and in the Matrigel plug assay. Unfortunately, the evidence shown seems entirely preliminary. Also, the study provides no evidence that these effects are important for angiogenesis during tumor progression.

There is already extensive evidence that sE-cad derived from proteolytic cleavage of membrane-bound E-cadherin facilitated tumor growth by activating, among others, the Pi3K/Akt pathway. The authors have cited publications (e.g. ref 38) showing that a treatment with anti-sE-cad antibodies reduces tumor growth. Therefore, the finding that blocking this molecule has anti-tumor effects is certainly not unexpected.

The authors show that conditioned medium derived from two ovarian carcinoma cell lines (Caov-3 and OV-9) stimulates HUVEC migration or tube formation that can be blocked, at least in part, by anti-sE-cad antibody HECD-1. Considering that VEGF is known to be abundantly released by ovarian cancer cells, it is a bit surprising that HECD-1 had such effects. However, the authors did not test an anti-VEGF antibody in the same assay. sE-cad, however, had no effect on proliferation, a key aspect of the angiogenesis cascade.

The authors also show that HECD-1 reduces the stimulation of endothelial cell migration induced by exosomes derived from Caov-3 and OV-9 cells stimulate endothelial cell migration. These findings suggest that exosomes indeed contain some sE-cad. The study however does not address the relative contribution of exosome-derived as opposed to shed sE-cad. Is it a minor or significant contribution?

My most significant concern is the complete lack of evidence that the reported pro-angiogenic effects of sE-cad, whether shed or exosome-derived, actually play a role in ovarian tumor angiogenesis. Documenting such effects would require much more extensive in vivo studies, with or without anti-VEGF agents.

Reviewer #2 (Remarks to the Author):

This paper describes a potential role for soluble E-cadherin as a stimulator of angiogenesis in ovarian cancer and demonstrates that it is released from ovarian cancer cells in the form of exosomes. I believe that this work is original but the paper could be improved with a number of additions/clarifications.

1. Throughout the paper the authors use the term 'ovarian cancer'. It is now accepted that ovarian cancer is in fact at least four different cancers that originate in different tissues, have distinct oncogenic mutations, different natural histories and response to treatment. Most researchers working on these cancers will now define more clearly the cancer(s) they are studying. In addition, great care must be taken when using ovarian cancer cell lines as most of them are not representative of the most common subtype high-grade serous ovarian cancer as was first

demonstrated in a paper published in Nature Communications in 2013 (Domcke et al). The impact and clarity of this paper is compromised by the fact that the authors ignore this. Of the three cell lines used in this paper, Domcke et al classified 2 as possibly serous ovarian cancer and one as likely serous ovarian cancer, yet the patient samples used in the study were a mix of 5 different cancers – serous, endometrioid, mucinous, clear cell and rare. As supplementary table 1 shows, the non-serous types of ovarian cancer seemed to generate more e-cadherin positive exosomes. This discrepancy should at least be acknowledged and discussed but work with cell lines that are representative of the other cancers would improve this paper.

2. As E-cadherin release is found in these different ovarian cancers is it still specific to these peritoneal cancers or do other cancer types have the same abilities to shed sE-cad? The authors note that circulating sE-cad is found in the blood of patients with several different cancers which would suggest that sE-cad is a common feature. It would increase the scope and importance of the paper if some data was included on other cancers. In particular, as exosomes can be obtained from human tissues, it would be interesting to know if sE-cad containing exosomes could be generated by ovarian and other common cancers.

3. As none of the ovarian cancers are thought to arise from the surface of the ovary (serous probably arises from the fallopian tube, endometrioid and clear cell from the uterus and mucinous from the bowel), ovarian surface epithelium are not the appropriate control cells to use.

4. The authors should include data to determine if VEGF synergises with sE-cad in the matrigel models and also if sE-cad induces VEGF in malignant cells especially in view of the signaling pathways identified. In addition, it is important to know if anti-VEGF treatment inhibits the angiogenesis stimulated by sE-cad.

5. In Figure 6, the number of mice in each experimental group should be given in the Figure legend.

Reviewer #3 (Remarks to the Author):

To authors

In this manuscript, the authors described the angiogenetic function of soluble E-cadherin on exosomes derived from ovarian cancer cells. The detail function and localization of exosomal sE-cad was investigated, and the effects of the exosomes were validated in in vivo experiments and clinical samples. Although the concept of this manuscript is interesting, there are several major concerns in this work as shown in following comments.

Major comments

1. It is difficult to understand that sE-cad truly localizes on surface on exosomes. The direct evidence of the fact was not found in this work. In Figure 2a, exosomes derived from Caov-3 and OV-90 cells should be included. In Figure 2d, the bands of 80 kDa (sE-cad) were not founded. If the full length of E-cad is cleaved in Golgi/TGN network, sE-cad can exist inside of exosomes. At least, immunoelectron microscopic analysis should be performed.

2. It is difficult to imagine the proposed pathway from secretion of sE-cad related exosomes to angiogenesis. As written in the above comment, localization of sE-cad is unclear, and how sE-cad bind to VE-cad is also unclear because sE-cad can be included inside exosomes (or just contained in exosome solution). The authors concluded that there are two independent pathways, and the proposed pathway map from secretion of sE-cad (+) exosomes to angiogenesis should be provided.

3. In the method section, the description of clinical samples is missing. All samples were approved by institutional ethical committees? The method of sample collection and exosome purification should be provided.

4. The particle number of all exosomes in this work is missing, and this information is crucial. There may be no correlation between protein level and particle number in clinical samples. As written in the method section, OV-90 exo. (25 ug per dose) and patient-derived exo. (15 ug per dose) were used in in vivo experiment. These amount of exosomes is very high. How many

number of exosome were used? How about the low dose assessment? How much was the original volume of culture supernatant and ascites needed for one mouse throughout the whole experiment? Please discuss about this point.

5. In page 14, "The expression levels of sE-cad-positive exosomes were tended to be higher in advanced stage tumors (stages III/IV) than in early stage tumors (stages I/II) (P = 0.1616) (Supplementary Table 1). High levels of sE-cad-positive exosomes also appeared to correlate with unfavorable survival outcome in the analysis (P = 0.215) (Fig. 6d)", it should be avoid to state these conclusions, because there were no significant differences. Furthermore, the number of stage I + II, in Table S1, 4 and 2 patients were too small to discuss.

6. It is interesting to know the level of exosomal sE-cad in healthy donors or non-cancerous patients, like benign tumors.

7. Did β -catenin include in exosomes? Only in endothelial cells? Please clarify this point.

8. In Figure 2d, it would be better to discuss the difference of Caov-3 exosomes and OV-90 exosomes. CTF-E-cad was abundant in Caov-3 exosomes, whereas FL-E-cad was abundant in OV-90 exosomes.

9. In Figure 2e, the images of exosomes were quite poor. They are not acceptable.

Other comments

1. In page 4, "Importantly, sE-cad is highly expressed in the serum and ascites of ovarian cancer (6.18-89.56 ug/ml)", it would be more informative for future research that sE-cad in serum exist in exosomes or not. Furthermore, it is better to show the concentration of sE-cad in ascites.

2. In page 5, line 11 "(Fig. S1)", the style indicating Figure number should be unify.

3. In page 5, line 13, "Caov-3 and OV-90, which had the highest sE-cad content of all cells" the term "all cells" is slightly exaggerative.

4. In the Result section, paragraph 2, the word, "chemotactic", does not fit in this case because this phenotype is only migration.

5. In page 5, line 18, "(Supplementary Fig. 2)." the comma should be removed.

6. In page 6, line 10-1, the microscopic images of the breakdown of the vascular barrier should be provided.

7. In page 7, "This model has been standardized extensively for studying morphological and functional neovascularization", adequate references should be needed.

8. In Figure 5a, the summarized network image is too small to see the names of genes. It should be revised.

9. In Figure 6d, the value of each N was absent. It should be included in its legend.

10. It is unclear about statistical analysis. Were all significant difference in bar charts analyzed by ANOVA?

11. In the several part of this manuscript, the terms "dose-dependent" or "time-dependent manner" appeared. However the trend was not fully described, because, for examples, in Figure 4d, there were no differences between the values of density at 30 min and 60 min. Were there significant differences between 30 min and 60 min?

12. In page 12, line 10, where is "Fig 7A"?

13. In vivo experiments, HAYA8 cells were used. Does this cell also secrete sE-cad.

14. In page 14, the definition of sE-cad level "high (+++), medium (++) , low (+), and none (-)" should be provided in detail.

Below we address the reviewers' comments point by point. Page, line, and figure number refer to the revised manuscript.

Reviewer #1

1. The authors have cited publications (e.g. ref 38) showing that a treatment with anti-sE-cad antibodies reduces tumor growth. Therefore, the finding that blocking this molecule has anti-tumor effects is certainly not unexpected.

Reply: The reviewer is correct that a treatment with anti-sE-cad reduces tumor growth. However, previous studies focused on the role of sE-cad in cell viability. It is still not known whether sE-cad affects angiogenesis, although this is a key mechanism for both primary tumor and metastatic growth. In ovarian cancer, angiogenesis is also associated with the formation of malignant ascites characteristic of tumor aggressiveness and metastatic potential. Current approved treatments are not effective in preventing ascitic fluid accumulation. Our findings are the first to demonstrate sE-cad's role in the pathogenesis of malignant ascites and reveal a new direction for understanding the oncogenic roles of sE-cad (p. 3, lines 5-7; p. 22, lines 11-14).

2. The authors show that conditioned medium derived from two ovarian carcinoma cell lines (Caov-3 and OV-9) stimulates HUVEC migration or tube formation that can be blocked, at least in part, by anti-sE-cad antibody HECD-1. Considering that VEGF is known to be abundantly released by ovarian cancer cells, it is a bit surprising that HECD-1 had such effects. However, the authors did not test an anti-VEGF antibody in the same assay. sE-cad, however, had no effect on proliferation, a key aspect of the angiogenesis cascade.

Reply: In response to the reviewer's suggestion, we have now tested an anti-VEGF antibody in the same assay. Consistent with previous observation (Ref.#14), Caov-3 and OV-90 express low levels of detectable VEGF (pg/mL) (Supplementary Fig. 5c) and the addition of anti-VEGF, at a concentration of 10 µg/mL which was shown to completely block the mitogenic activity of 10 ng/mL VEGF in HUVEC (Ref.#15), did not cause significant alterations of conditioned media or sE-cad-positive exosomes induced endothelial cell migration and tube formation (Supplementary Fig. 5d, 5e, and 8a) (p. 7, lines 11-16; p. 11, lines 9-12). The observation that sE-cad is a non-mitogenic angiogenesis factor is consistent with many cell adhesion molecules that can mediate angiogenesis independent of any mitogenic effect in the literature (e.g. Ref.#26, 27). Although the angiogenic effect is not associated with proliferation, these findings can be interpreted in light of their potent migration potential, which is shown to be most closely associated with the angiogenic function (Ref.#28) (p. 18, lines 6-11).

3. The authors also show that HECD-1 reduces the stimulation of endothelial cell migration

induced by exosomes derived from Caov-3 and OV-9 cells stimulate endothelial cell migration. These findings suggest that exosomes indeed contain some sE-cad. The study however does not address the relative contribution of exosome-derived as opposed to shed sE-cad. Is it a minor or significant contribution?

Reply: Following the reviewer's suggestion, we examined the relative contribution of exosome-derived as opposed to shed sE-cad on endothelial cell migration and on tube formation and showed that sE-cad-positive exosomes demonstrated a similar angiogenic response even at a protein concentration 4 times lower than shed sE-cad, suggesting that E-cad-positive exosomes are more potent in promoting these angiogenic effects (Supplementary Fig. 7) (p.11, lines 6-9). These findings, consistent with the predominant (~70%) expression of E-cadherin in Golgi/Trans-Golgi that we observe (Fig. 2b), suggest that exosome-derived sE-cad is major contributor to angiogenesis.

4. My most significant concern is the complete lack of evidence that the reported pro-angiogenic effects of sE-cad, whether shed or exosome-derived, actually play a role in ovarian tumor angiogenesis. Documenting such effects would require much more extensive in vivo studies, with or without anti-VEGF agents.

Reply: As we replied above, we have now tested an anti-VEGF antibody in the same assay. The addition of anti-VEGF, at a concentration of 10 µg/mL which was shown to completely block the mitogenic activity of 10 ng/mL VEGF in HUVEC (Ref.#15), did not cause significant alterations of conditioned media or sE-cad-positive exosomes induced endothelial cell migration and tube formation (Supplementary Fig. 5d, 5e, and 8a) (p. 7, lines 11-16; p. 11, lines 9-12). We also showed that sE-cad had no effect on VEGF expression (Supplementary Fig. 8b). VEGF synergized with sE-cad in the angiogenic response (Supplementary Fig. 8c) (p. 11, lines 12-18, p. 12, line 1). Because these effects were not mediated through VEGF, we did not evaluate anti-VEGF further in vivo. Nevertheless, to further demonstrate the pro-angiogenic effects of sE-cad in vivo, we have added data on microvessel density and vascular leakage assay (Fig. 6b and c) (p. 16, lines 4-8), which are standard and widely used assays for in vivo angiogenesis. The findings, in keeping with the data on in vitro angiogenic activities (Fig. 1), clearly support our conclusion for an angiogenic role of sE-cad.

Reviewer #2

1. The impact and clarity of this paper is compromised by the fact that the authors ignore this. Of the three cell lines used in this paper, Domcke et al classified 2 as possibly serous ovarian cancer and one as likely serous ovarian cancer, yet the patient samples used in the study were a mix of 5 different cancers – serous, endometrioid, mucinous, clear cell and rare. As supplementary table 1 shows, the non-serous types of ovarian cancer seemed to

generate more e-cadherin positive exosomes. This discrepancy should at least be acknowledged and discussed but work with cell lines that are representative of the other cancers would improve this paper.

Reply: The reviewer notes the importance of the choice of cell lines. As suggested, we now included the use of OVK18, TOV112D (endometrioid), TOV21G (clear cell), and MCAS (mucinous) ovarian cancer cells. sE-cad can only be found in MCAS (Supplementary Fig. 9) (p. 17, lines 1-2). Although not requested by the reviewer, we have added data (7 cases) on the patients' samples (new total 35) (18 serous, 3 endometrioid, 4 mucinous, 6 clear cell, and 4 rare subtypes). We found no association between strong sE-cad-positive exosomes and histologic subtype ($P = 0.7107$) (p. 16, lines 17-18; p. 17, line 1). The relatively small number of patients may have biased the earlier difference (Supplementary Table 1).

2. As E-cadherin release is found in these different ovarian cancers is it still specific to these peritoneal cancers or do other cancer types have the same abilities to shed sE-cad? The authors note that circulating sE-cad is found in the blood of patients with several different cancers which would suggest that sE-cad is a common feature. It would increase the scope and importance of the paper if some data was included on other cancers. In particular, as exosomes can be obtained from human tissues, it would be interesting to know if sE-cad containing exosomes could be generated by ovarian and other common cancers.

Reply: As suggested, we also performed experiments which showed that sE-cad containing exosomes were present and were able to induce angiogenesis in MCF-7 breast and HCT116 colon cancer cells (Supplementary Fig. 11), indicating that the angiogenic function of sE-cad that we propose in ovarian cancer may have broad implications for other tumor cell types. Moreover, sE-cad-positive exosomes can also be found in the ascitic fluid from patients with colon, breast, liver, endometrial, and stomach cancer (Supplementary Table 2) (p. 22, lines 16-18; p. 23, lines 1-2).

3. As none of the ovarian cancers are thought to arise from the surface of the ovary (serous probably arises from the fallopian tube, endometrioid and clear cell from the uterus and mucinous from the bowel), ovarian surface epithelium are not the appropriate control cells to use.

Reply: As suggested, we have tested whether the expression of sE-cad on normal fallopian tube epithelial (FTE) cells as a control. There was no obvious sE-cad expression in these cells, demonstrating that the expression of sE-cad is specific to ovarian cancer cells (Supplementary Fig. 1b) (p. 5, line 13).

4. The authors should include data to determine if VEGF synergises with sE-cad in the

matrigel models and also if sE-cad induces VEGF in malignant cells especially in view of the signaling pathways identified. In addition, it is important to know if anti-VEGF treatment inhibits the angiogenesis stimulated by sE-cad.

Reply: The addition of anti-VEGF, at a concentration of 10 µg/mL which was shown to completely block the mitogenic activity of 10 ng/mL VEGF in HUVEC (Ref.#15), did not cause significant alterations of conditioned media or sE-cad-positive exosomes induced endothelial cell migration and tube formation (Supplementary Fig. 5d, 5e, and 8a) (p. 7, lines 11-16; ; p. 11, lines 9-12). We have evaluated the impact of sE-cad on VEGF expression and found no effect (Supplementary Fig. 8b). We have also provided evidence that VEGF synergized with sE-cad in the endothelial tube formation in Matrigel (Supplementary Fig. 8c) (p. 11, lines 12-18, p. 12, line 1). Because these effects are not mediated through VEGF, we suggest that the angiogenic effect of sE-cad is independent of VEGF, and presumably the existence of alternative proangiogenic factors and signaling molecules are involved (p. 22, lines 1-2).

5. In Figure 6, the number of mice in each experimental group should be given in the Figure legend.

Reply: The number of mice in each experimental group is now described.

Reviewer #3

1. It is difficult to understand that sE-cad truly localizes on surface on exosomes. The direct evidence of the fact was not found in this work. In Figure 2a, exosomes derived from Caov-3 and OV-90 cells should be included. In Figure 2d, the bands of 80 kDa (sE-cad) were not founded. If the full length of E-cad is cleaved in Golgi/TGN network, sE-cad can exist inside of exosomes. At least, immunoelectron microscopic analysis should be performed.

Reply: As suggested, we performed the immunoelectron microscopic analysis to investigate sE-cad localization. Gold particles were detected on the exosomal surface using the anti-E-cadherin antibody (HECD-1). Similarly, exosomes were positive for the exosomal marker tetraspan (CD63) on their surface. No staining was detected on exosomes stained with control antibody, confirming the cell surface localization of sE-cad (Fig. 2e) (p. 9, lines 17-18; p. 10, lines 1-2; p. 28, line 17-18; p. 29, line 1).

2. It is difficult to imagine the proposed pathway from secretion of sE-cad related exosomes to angiogenesis. As written in the above comment, localization of sE-cad is unclear, and how sE-cad bind to VE-cad is also unclear because sE-cad can be included inside exosomes (or just contained in exosome solution). The authors concluded that there are two independent pathways, and the proposed pathway map from secretion of sE-cad (+)

exosomes to angiogenesis should be provided.

Reply: We examined the internalization of labeled purified exosomes by fluorescent microscopy. Exosomes were labeled with a fluorescent BODIPY-TR ceramide. Unbound dye was removed using exosome spin columns. In addition, to control for incomplete dye removal, we columned purified dye in the absence of sE-cad-positive exosomes. Labeled exosomes were added to HUVECs. We showed that the labeling occurred on HUVECs, which could be reverted by treatment with HECD-1, suggesting the uptake of exogenous purified exosomes by HUVECs. In contrast, no labeling was observed in the control sample (Fig. 3e) (p. 10, lines 14-18; p. 29, lines 8-17).

3. In the method section, the description of clinical samples is missing. All samples were approved by institutional ethical committees? The method of sample collection and exosome purification should be provided.

Reply: The method of sample collection and exosome purification are now provided in sections “Cell Lines, Patient Samples, and Cell Culture” and “Exosome Isolation from Ascitic Fluid” under METHODS (p.24, lines 2-10; p. 28, lines 6-7).

4. The particle number of all exosomes in this work is missing, and this information is crucial. There may be no correlation between protein level and particle number in clinical samples. As written in the method section, OV-90 exo. (25 ug per dose) and patient-derived exo. (15 ug per dose) were used in in vivo experiment. These amount of exosomes is very high. How many number of exosome were used? How about the low dose assessment? How much was the original volume of culture supernatant and ascites needed for one mouse throughout the whole experiment? Please discuss about this point.

Reply: The particle number of exosomes is measured by ZetaView, and there are 4.1×10^7 particles per μg of OV-90 exosomes and 2.4×10^7 particles per μg of Caov-3 exosomes (Fig. 2f) (p. 10, lines 2-5). Low dose assessment is now performed by comparing the angiogenic effect of 5 μg , 15 μg and 25 μg of exosomes (Supplementary Fig. 6e). For each mouse, around 75 mL culture supernatant or 25 mL ascites is needed (p. 15, lines 15-17).

5. In page 14, “The expression levels of sE-cad-positive exosomes were tended to be higher in advanced stage tumors (stages III/IV) than in early stage tumors (stages I/II) (P = 0.1616) (Supplementary Table 1). High levels of sE-cad-positive exosomes also appeared to correlate with unfavorable survival outcome in the analysis (P = 0.215) (Fig. 6d)”, it should be avoid to state these conclusions, because there were no significant differences. Furthermore, the number of stage I + II, in Table S1, 4 and 2 patients were too small to discuss.

Reply: Although not requested, we have added data on the patients' samples (new total 35) (18 serous, 3 endometrioid, 4 mucinous, 6 clear cell and 4 rare subtypes). However, we note the still relatively small sample size. In the revised manuscript, we also tuned down our claim and delete the conclusion that "high levels of sE-cad-positive exosomes also appeared to correlate with unfavorable survival outcome in the analysis" as suggested (p. 16, lines 15-18; p. 17, lines 1-4).

6. It is interesting to know the level of exosomal sE-cad in healthy donors or non-cancerous patients, like benign tumors.

Reply: Following this suggestion, we have performed the experiments on 6 benign cases. Our results showed that there was no or little (3.92 – 8.13 $\mu\text{g/mL}$) exosomal sE-cad in the peritoneal fluids of these patients (Supplementary Table 2) (p. 17, lines 2-4).

7. Did β -catenin include in exosomes? Only in endothelial cells? Please clarify this point.

Reply: We and others have shown that β -catenin protein was included in exosomes (Supplementary Fig. 10) (Ref.#41), thus it is possible that β -catenin may come from an exogenous source (p. 20, lines 14-18).

8. In Figure 2d, it would be better to discuss the difference of Caov-3 exosomes and OV-90 exosomes. CTF-E-cad was abundant in Caov-3 exosomes, whereas FL-E-cad was abundant in OV-90 exosomes.

Reply: How sE-cad is cleaved is currently under investigation. Nevertheless, our results showed that OV-90 which expressed high levels of FL-E-cad showed much increased responses than in Caov-3, suggesting that FL-E-cad on exosomes is likely to be responsible for the angiogenic effect (Fig. 3d and Supplementary Fig. 6c) (p. 11, lines 2-4).

9. In Figure 2e, the images of exosomes were quite poor. They are not acceptable.

Reply: Fig. 2e is replaced with high resolution images of exosomes.

Other comments

1. In page 4, "Importantly, sE-cad is highly expressed in the serum and ascites of ovarian cancer (6.18-89.56 $\mu\text{g/ml}$)", it would be more informative for future research that sE-cad in serum exist in exosomes or not. Furthermore, it is better to show the concentration of sE-cad in ascites.

Reply: Whether sE-cad-positive exosomes are present in the serum of ovarian cancer patients is surely a direction we would like to pursue in the future. The concentration of sE-cad-positive exosomes in ascites in our analysis (2.62 – 27.10 $\mu\text{g/mL}$) is stated (p.16, lines 15-17).

2. In page 5, line 11 "(Fig. S1)", the style indicating Figure number should be unify.
Reply: The style is unified as suggested (p. 5, line 11).
3. In page 5, line 13, "Caov-3 and OV-90, which had the highest sE-cad content of all cells" the term "all cells" is slightly exaggerative.
Reply: The sentence has been revised to replace 'all' with 'these' to make it clear (p.5, lines 13-15).
4. In the Result section, paragraph 2, the word, "chemotactic", does not fit in this case because this phenotype is only migration.
Reply: The word 'chemotactic' is deleted (p.5, line 17; p. 6, line 4).
5. In page 5, line 18, "(Supplementary Fig. 2)." the comma should be removed.
Reply: The comma has been removed (p. 5, line 18).
6. In page 6, line 10-1, the microscopic images of the breakdown of the vascular barrier should be provided.
Reply: It is known that the breakdown of the vascular barrier has an important role in ascites formation and enhances metastasis. We have revised the sentence and included a reference (#12). The barrier function is measured by the FITC-dextran flux (p. 6, lines 9-10).
7. In page 7, "This model has been standardized extensively for studying morphological and functional neovascularization", adequate references should be needed.
Reply: The references (#16) have been added (p. 8, lines 1-2).
8. In Figure 5a, the summarized network image is too small to see the names of genes. It should be revised.
Reply: The font size of the images has been revised to make it clear.
9. In Figure 6d, the value of each N was absent. It should be included in its legend.
Reply: The value of each N is now included (p. 53, line 6).
10. It is unclear about statistical analysis. Were all significant difference in bar charts analyzed by ANOVA?
Reply: The section is now revised to make it clear (p. 35, lines 10-13).

11. In the several part of this manuscript, the terms “dose-dependent” or “time-dependent manner” appeared. However the trend was not fully described, because, for examples, in Figure 4d, there were no differences between the values of density at 30 min and 60 min. Were there significant differences between 30 min and 60 min?

Reply: There was no significant difference between 30 min and 60 min. We have revised the sentence to make it clear (p. 13, line 6).

12. In page 12, line 10, where is “Fig 7A”?

Reply: The revision is made (Fig. 5a) (p.14, line 3).

13. In vivo experiments, HAYA8 cells were used. Does this cell also secrete sE-cad.

Reply: HEYA8 cells do not secrete sE-cad (Supplementary Fig. 1c) (p. 15, line 13).

14. In page 14, the definition of sE-cad level “high (+++), medium (++), low (+), and none (-)” should be provided in detail.

Reply: sE-cad level “high (+++), medium (++), low (+), and none (-)” refer to $> 20 \mu\text{g/mL}$, $10 - 20 \mu\text{g/mL}$, $< 10 \mu\text{g/mL}$ and no expression respectively and this information is now included in the RESULTS and METHODS sections (p. 16, lines 14-15; p. 28, lines 11-13).

Reviewers' comments:

Reviewer #1 (Remarks to the Author):

The authors have made an effort to address my comments. However, some of their responses are entirely unconvincing. In response to my request to test their reagents with and without VEGF inhibitors (an FDA-approved treatment for ovarian cancer), they answer that VEGF is expressed at low levels in the conditioned media of CAOV3 cells and therefore plays no role in the angiogenesis induced by such cells. This claim is contradicted by numerous papers published over the last twenty years documenting the profound effects of VEGF inhibition on angiogenesis and ascites formation following implantation of CAOV3 and other ovarian tumor cell lines in immunodeficient mice. See, for example, Mesiano et al, *Am J Pathol*, 153:1249-56, 1998; Byrne et al, *Clin Cancer Res* 9:5721-28, 2003; Mabuchi et al, *Clin Cancer Res.*, 14:7781-9, 2008; Huang et al, *Mol Cancer Ther* 15:1344-52, 2016. The authors did not cite any of these papers, although they are well-known and well-cited in the field of ovarian cancer/angiogenesis. The authors' claim that there is no need to test VEGF inhibition in vivo in this model because this molecule is not implicated in angiogenesis in the model examined has simply no plausibility.

Other experiments lack some critical controls. For example, In Supplementary Fig 5, the authors report that an anti-VEGF antibody fails to block the effect the conditioned on endothelial cells. However, they provide no evidence that the antibody neutralizes the effects of VEGF.

Overall I find this revision even less convincing than the original submission.

Reviewer #2 (Remarks to the Author):

The reviewers raised an extensive list of concerns about the work in this paper. The authors have conducted new experiments that address these concerns. The authors have addressed the concerns I raised in a satisfactory manner.

Minor point - one of the sentences in the Introduction lines (53-56) does not make sense.

Reviewer #3 (Remarks to the Author):

This revised version is much improved. The authors addressed almost all of my concerns.

Re: NCOMMS-17-07045B “Soluble E-cadherin promotes tumor angiogenesis and localizes to exosome surface”

We sincerely thank the reviewers for their helpful comments. In response to Reviewers' comments, we have carried out additional experiment that we think further improve the manuscript.

Page, line, and figure numbers refer to the revised manuscript. Changes are highlighted.

Reviewer #1

1. In response to my request to test their reagents with and without VEGF inhibitors (an FDA-approved treatment for ovarian cancer), they answer that VEGF is expressed at low levels in the conditioned media of CAOV3 cells and therefore plays no role in the angiogenesis induced by such cells. This claim is contradicted by numerous papers published over the last twenty years documenting the profound effects of VEGF inhibition on angiogenesis and ascites formation following implantation of CAOV3 and other ovarian tumor cell lines in immunodeficient mice. See, for example, Mesiano et al, Am J Pathol, 153:1249-56, 1998; Byrne et al, Clin Cancer Res 9:5721-28, 2003; Mabuchi et al, Clin Cancer Res., 14:7781-9, 2008; Huang et al, Mol Cancer Ther 15:1344-52, 2016. The authors did not cite any of these papers, although they are well-known and well-cited in the field of ovarian cancer/angiogenesis. The authors' claim that there is no need to test VEGF inhibition in vivo in this model because this molecule is not implicated in angiogenesis in the model examined has simply no plausibility.

Reply: The reviewer is correct that the profound effects of VEGF inhibition on angiogenesis and ascites formation following implantation of CaOV3 and other ovarian tumor cell lines (including OVCAR-3 and SKOV-3) in immunodeficient mice are demonstrated. The references are cited (Ref. #46, 47, 49 and 50). However, these experiments are carried out in vivo. We suspect that there are two reasons for the lack of such angiogenic responses in culture that we observe. First are the very low levels of VEGF achieved in vitro. The little or no activities of Caov-3 conditioned media in endothelial cell migration and tube formation actually correlates well with the small amounts of VEGF secreted by these cells (CaOV3<OVCAR-3<<SKOV-3) (CaOV-3 in pg/ml) (Ptaszynska et al., 2008; Sher et al., 2009; Bourgeois et al., 2015). 2 ng/ml VEGF is a sub-threshold dose that induces no effect on endothelial cell tube formation (Supplementary Figure 8c), in corroboration with previous observations (Bai et al., 2014). Second, while such an effect was not demonstrated in culture, it is probable that, in vivo, VEGF may reach levels that promote ascites formation. VEGF release requires cleavage by serum and ascites proteases (Ref. #17). Our culture condition may be suboptimal for efficient cleavage (p. 7,

line 18 and p. 8, lines 1-3). The findings that the addition of anti-VEGF did not cause significant alterations of conditioned media or sE-cad-positive exosomes induced endothelial cell migration and tube formation (Supplementary Fig. 8a), sE-cad-positive exosomes had no effect on VEGF expression (Supplementary Fig. 8b), and VEGF synergized with sE-cad-positive in the angiogenic response (Supplementary Fig. 8c) suggest that while both VEGF and sE-cad are important contributors to angiogenesis, they are independent pathways (p. 11, lines 14-18 and p. 12, lines 1-9).

2. In Supplementary Fig 5, the authors report that an anti-VEGF antibody fails to block the effect the conditioned on endothelial cells. However, they provide no evidence that the antibody neutralizes the effects of VEGF.

Reply: We did test the neutralizing sera which demonstrated that the endothelial tube formation of VEGF was reversed by anti-VEGF antibody treatment (Supplementary Fig. 8a). In response to the reviewer's suggestion, we have now tested the immunodepletion of VEGF from the conditioned media by anti-VEGF antibody (Supplementary Fig. 5d). The observation that VEGF could be selectively removed from the conditioned media with the anti-VEGF antibody is also consistent with that in the literature (e.g. Ref. #15).

Reviewer #2

1. One of the sentences in the Introduction lines (53-56) does not make sense.

Reply: The sentence is now revised to make it clear (p. 3, line 18 and p. 4, lines 1-3).

We have also corrected some minor mistakes that we found during the revision, and they are also highlighted in the mark-up version of the manuscript. Again, many thanks for allowing us to revise the paper. We hope that this manuscript will now be acceptable for publication.